METHODS AND RESOURCES

# A sensitive and specific genetically-encoded potassium ion biosensor for in vivo applications across the tree of life

**Sheng-Yi Wu[1], Yurong Wen[2,3], Nelson B. C. Serre[4], Cathrine Charlotte Heiede Laursen[5], Andrea Grostøl Dietz[5], Brian R. Taylor[6], Mikhail Drobizhev[7], Rosana S. Molina[7], Abhi Aggarwal[8], Vladimir Rancic[9], Michael Becker[10], Klaus Ballanyi[9], Kaspar Podgorski[8], Hajime Hirase[5], Maiken Nedergaard[5,11], Matyáš Fendrych[4], M. Joanne Lemieux[2], Daniel F. Eberl[12], Alan R. Kay[12], Robert E. Campbell[1,13]\*, Yi Shen[1]\***

**1** Department of Chemistry, University of Alberta, Edmonton, Alberta, Canada, **2** Department of Biochemistry, University of Alberta, Edmonton, Alberta, Canada, **3** Center for Microbiome Research of Med-X Institute, The First Affiliated Hospital, Xi'an Jiaotong University, Xi'an, Shaanxi, China, **4** Department of Experimental Plant Biology, Charles University, Prague, Czech Republic, **5** Center for Translational Neuromedicine, University of Copenhagen, Copenhagen, Denmark, **6** Department of Physics, University of California at San Diego, La Jolla, California, United States of America, **7** Department of Microbiology and Cell Biology, Montana State University, Bozeman, Montana, United States of America, **8** Janelia Research Campus, Howard Hughes Medical Institute, Ashburn, Virginia, United States of America, **9** Department of Physiology, University of Alberta, Edmonton, Alberta, Canada, **10** GM/CA@APS, X-ray Science Division, Advanced Photon Source, Argonne National Laboratory, Argonne, Illinois, United States of America, **11** Center for Translational Neuromedicine, University of Rochester Medical Center, Rochester, New York, United States of America, **12** Department of Biology, University of Iowa, Iowa City, Iowa, United States of America, **13** Department of Chemistry, The University of Tokyo, Tokyo, Japan

\* robert.e.campbell@ualberta.ca (REC); yi.shen@ualberta.ca (YS)

**Data Availability Statement:** Data availability Plasmids and DNA sequences are available via Addgene (Addgene ID 177116, 177117). The GINKO1 structure is deposited in the Protein Data

## Abstract

Potassium ion ($K^+$) plays a critical role as an essential electrolyte in all biological systems. Genetically-encoded fluorescent $K^+$ biosensors are promising tools to further improve our understanding of $K^+$-dependent processes under normal and pathological conditions. Here, we report the crystal structure of a previously reported genetically-encoded fluorescent $K^+$ biosensor, GINKO1, in the $K^+$-bound state. Using structure-guided optimization and directed evolution, we have engineered an improved $K^+$ biosensor, designated GINKO2, with higher sensitivity and specificity. We have demonstrated the utility of GINKO2 for in vivo detection and imaging of $K^+$ dynamics in multiple model organisms, including bacteria, plants, and mice.

## Introduction

The potassium ion ($K^+$) is one of the most abundant cations across biological systems [1]. It is involved in a variety of cellular activities in organisms ranging from prokaryotes to multicellular eukaryotes [2–4]. While studies of other biologically important cations, notably calcium ion ($Ca^{2+}$), have been revolutionized by the availability of high-performance genetically-encoded biosensors [5,6], the development of analogous biosensors for $K^+$ has lagged far behind. Canonical methods to monitor $K^+$ include $K^+$-sensitive microelectrodes and synthetic

Bank (PDB ID:7VCM). Seeds are deposited in the NASC Arabidopsis seed repository (https://arabidopsis.info/ ID: N2111001). Data supporting the findings in this research are included in the supplementary data file. Code availability Root elongation quantification code is available at https://sourceforge.net/projects/lbopsis/.

**Funding:** This work was supported by grants from the Canadian Institutes of Health Research (CIHR, FS-154310 to REC) and the Natural Sciences and Engineering Research Council of Canada (NSERC, RGPIN 2018-04364 to REC, RGPIN-2020-05514 to KB, and RGPIN-2016-06478 to MJL). SYW was supported by NSERC Canada Graduate Scholarships – Doctoral program, Alberta Innovates Technology Future (AITF) Graduate Scholarship, and the University of Alberta. YW was supported by the Alberta Parkinson Society Fellowship and National Natural Science Foundation of China (No. 31870132 and No. 82072237). This research used resources of the Advanced Photon Source (APS), a U.S. Department of Energy (DOE) Office of Science User Facility operated for the DOE Office of Science by Argonne National Laboratory under Contract No. DE-AC02-06CH11357. X-ray crystallography was performed using Beamline 23IDB at APS. GM/CA@APS has been funded by the National Cancer Institute (ACB-12002) and the National Institute of General Medical Sciences (AGM-12006, P30GM138396). Data were also collected at beamline CMCF-ID at the Canadian Light Source, a national research facility of the University of Saskatchewan, which is supported by the Canada Foundation for Innovation (CFI), the NSERC, the National Research Council (NRC), the CIHR, the Government of Saskatchewan, and the University of Saskatchewan. NBCS and MF work was supported by the European Research Council (Grant No. 803048) and Charles University Primus (Grant No. PRIMUS/19/SCI/09). CCHL, AGD, HH, and MN were supported by the Novo Nordisk Foundation (NNF19OC0058058 to HH and NNF13OC0004258 to MN) and the Lundbeck Foundation (R155-2016-552 to MN and R263-2017-4062 to AGD). BRT was supported by NIH grant R01GM095903. AA and KP were supported by Howard Hughes Medical Institute. ARK and DFE were supported in part by NSF grant 2037828. Two-photon characterization work (MD and RSM) was supported by the NIH BRAIN grant U24 NS109107 (Resource for Multiphoton Characterization of Genetically-Encoded Probes). The funders had no role in study design, data collection and analysis, decision to publish, or preparation of the manuscript.

dyes. Microelectrodes are considered the gold standard for their sensitivity and selectivity, but they are invasive and not suitable for high-throughput cellular or subcellular $K^+$ detection [7]. Synthetic dye-based approaches allow $K^+$ visualization in live cell populations with improved spatiotemporal resolution [8–11]; however, they still require dye loading and washing procedures and lack the targetability to specific cell types or subcellular compartments.

A high-performance genetically-encoded fluorescent biosensor for $K^+$ could enable a variety of applications that are currently impractical or impossible by enabling targeted expression and noninvasive in vivo imaging. We have previously reported a prototype intensiometric $K^+$ biosensor, designated GINKO1, based on the insertion of $K^+$-binding protein (Kbp) [12] into enhanced green fluorescent protein (EGFP) (**Fig 1A**) [13]. Ratiometric genetically-encoded biosensors have also been reported [13,14]. To create a more robust $K^+$ biosensor with broader utility, we undertook an effort to further improve the sensitivity and specificity of GINKO1.

## Results and discussion

### Structure of GINKO1

To better understand the $K^+$-dependent fluorescence response mechanism of GINKO1 and facilitate further engineering, we determined the crystallographic structure of GINKO1 in the $K^+$-bound state at 1.85 Å (**Figs 1B** and **S1** and **S1 Table**). Well-diffracting crystals of the unbound state were unattainable. The $K^+$-bound crystal structure revealed the location and coordination geometry of the $K^+$-binding site of Kbp (**Fig 1C**), which was not apparent from the previously reported NMR structure (**Fig 1D and 1E**) [12]. Notably, the $K^+$ ion is coordinated via 6 backbone carbonyl oxygen atoms (from amino acids V154, K155, A157, G222, I224, and I227). This coordination via backbone carbonyl oxygen atoms is similar to that observed in the $K^+$ selective filters of KcsA (PDB ID: 1BL8) [15] and TrkH (PDB ID: 4J9U) [16], as well as $K^+$-coordinating compound valinomycin [17]. The distances of coordinating carbonyl oxygens to $K^+$ in GINKO1 range from 2.6 to 3.2 Å with a mean value of 2.8 Å (**Fig 1C**), similar to those in KscA (2.70 to 3.08 Å, with a mean value of 2.85 Å) [18], and valinomycin (2.74 to 2.85 Å) [17]. One difference is that $K^+$ is coordinated via 8 oxygens from backbone carbonyls in both KcsA and TrkH, and 6 backbone carbonyls in Kbp.

In the previous study that described the Kbp NMR solution structure, it was suggested that crystallization of Kbp for X-ray crystallography was challenging [12]. We suspect that fusing Kbp to EGFP constrains the conformational mobility of Kbp, thus increasing the stability of Kbp protein for it to be crystallized as a domain in GINKO1. A similar approach has recently been reported to stabilize small transmembrane proteins for crystallization [19]. The Kbp region of the GINKO1 structure aligns well with the previous Kbp NMR solution structure (**Fig 1D and 1E**). The BON domain and the LysM domain of Kbp were both well resolved in the GINKO1 structure. The structure further revealed that the $K^+$ binding site is located in the BON domain, close to the interface between the BON and LysM domains (**Fig 1D**). This is consistent with the previous finding that the BON domain binds $K^+$ and the LysM domain stabilizes the $K^+$-bound BON domain [12].

### Engineering of GINKO2

Structure-guided mutagenesis and directed evolution were used to optimize GINKO1. Aligning structures of GINKO1 and GCaMP6 (**Fig 2A**) [20] revealed that GINKO1 E295 structurally aligns with GCaMP6 R376. R376 is engaged in a water-mediated interaction with the chromophore in GCaMP, likely acting to communicate the $Ca^{2+}$-dependent conformational change in the $Ca^{2+}$-binding domain to the GFP chromophore [21]. We mutated GINKO1 E295 to basic and hydrophobic amino acids (K/R/W/Y/P/L/F), with the hypothesis that these

**Competing interests:** The authors have declared that no competing interests exist.

**Abbreviations:** aCSF, artificial cerebrospinal fluid; CSD, cortical spreading depolarization; DMEM, Dulbecco's Modified Eagle Medium; EC, extinction coefficient; EGFP, enhanced green fluorescent protein; EP-PCR, error-prone PCR; FBS, fetal bovine serum; Kbp, K$^+$-binding protein; NMDG, N-methyl-D-glucamine; OD, optical density; QY, quantum yield; ROI, region of interest; 2P, 2-photon.

residues could similarly modulate the chromophore environment by introducing an opposite charge or removing the charge altogether. Among the E295 variants, E295F was selected as GINKO1.1 due to it having the largest K$^+$-dependent intensity change ($\Delta F/F_0 = 2.0$) (S2 Fig). As previous structural and mechanistic analysis of high-performance biosensors has suggested that the linker regions are of particular importance for biosensor function [22], we performed site-directed saturation mutagenesis on the linker residues connecting EGFP to Kbp and screened for variants with a larger $\Delta F/F_0$ (Fig 2B). This yielded GINKO1.2 with a linker sequence of A296-A297-N298 (Fig 2C) and a 30% improvement in $\Delta F/F_0$. We further optimized GINKO1.2 via directed evolution in *Escherichia coli* (S3 Fig). After multiple rounds of iterative evolution, we settled on a final variant, designated GINKO2, with substantially improved brightness and K$^+$ response (Figs 2D and S4 and S2 Table).

With the structural insight provided by the GINKO1 crystal structure, we were able to rationalize some critical mutations accumulated during GINKO2 engineering (S2 Table). K356R, a mutation that doubled the fluorescence change $\Delta F/F_0$, is located at the interface of the Kbp and EGFP in the GINKO1 structure (S4A Fig). This mutation may help to stabilize the K$^+$-bound GINKO1 by reducing the distance to D148 and hence increasing their electrostatic interaction (S4B Fig). Another case is mutation of a pair of lysines (K259N on Kbp domain and K102E on EGFP domain) that are in relatively close proximity (S4C Fig). The K259N and K102E mutations first appeared in 2 different variants in the GINKO1.5 library and provided only small improvements. When using both variants as templates to generate the GINKO1.6 library, K259N and K102E were simultaneously incorporated into the selected GINKO1.6.15 variant (S2 Table), which provided a substantial improvement of $\Delta F/F_0$ from approximately 2.5 to 3.5. The double mutation may help to further stabilize the interaction between Kbp and EGFP in the K$^+$-bound state (S4D Fig).

## Characterization of GINKO2

To characterize GINKO2 in vitro, we determined its fluorescence spectra, brightness, affinity, fluorescence change ($\Delta F/F_0$), specificity, kinetics, and pH dependence. Upon K$^+$ binding,

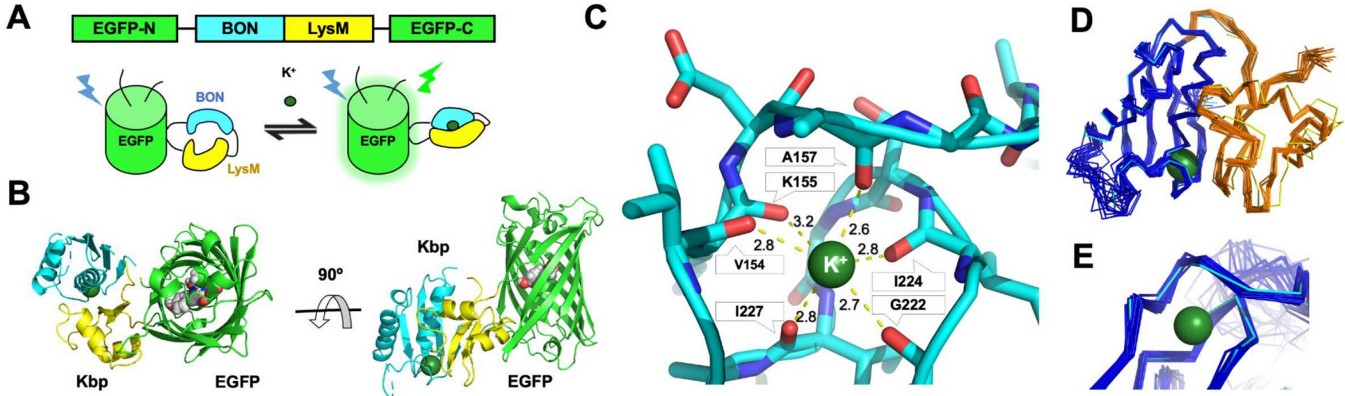

**Fig 1. GINKO1 structure.** (A) Schematic representation of GINKO. In the top panel, the linear DNA representation of GINKO gene shows the ligand recognition domain Kbp (BON in cyan and LysM in yellow) inserted in the split EGFP (green). In the bottom panel, the illustration shows a K$^+$-binding induced conformational change of Kbp leading a change in fluorescence. (B) Cartoon representation of the structure of GINKO1 with the BON (bacterial OsmY and nodulation) domain of Kbp in cyan, the LysM (lysin motif) domain of Kbp in yellow, and the EGFP in green. The chromophore and the K$^+$ ion (green) are shown in sphere representation. (C) The K$^+$ is coordinated by carbonyl backbone atoms of 6 amino acids. The distance (in Å) of each amino acid backbone oxygen to the K$^+$ ion was measured in PyMOL. (D) Structure alignment of the Kbp domain in GINKO1 and the previously reported NMR structure of Kbp (PDB ID: 5FIM). Kbp NMR structure ensemble is shown in ribbon representation. GINKO1 BON domain is in cyan; GINKO1 LysM domain is in yellow. Kbp NMR structure BON domain is in blue, and LysM domain is in orange. (E) Zoom-in view of the binding pocket in the GINKO1 crystal structure and the Kbp NMR structure (PDB ID: 5FIM). Structure coloring is the same as in (D). EGFP, enhanced green fluorescent protein; Kbp, K$^+$-binding protein.

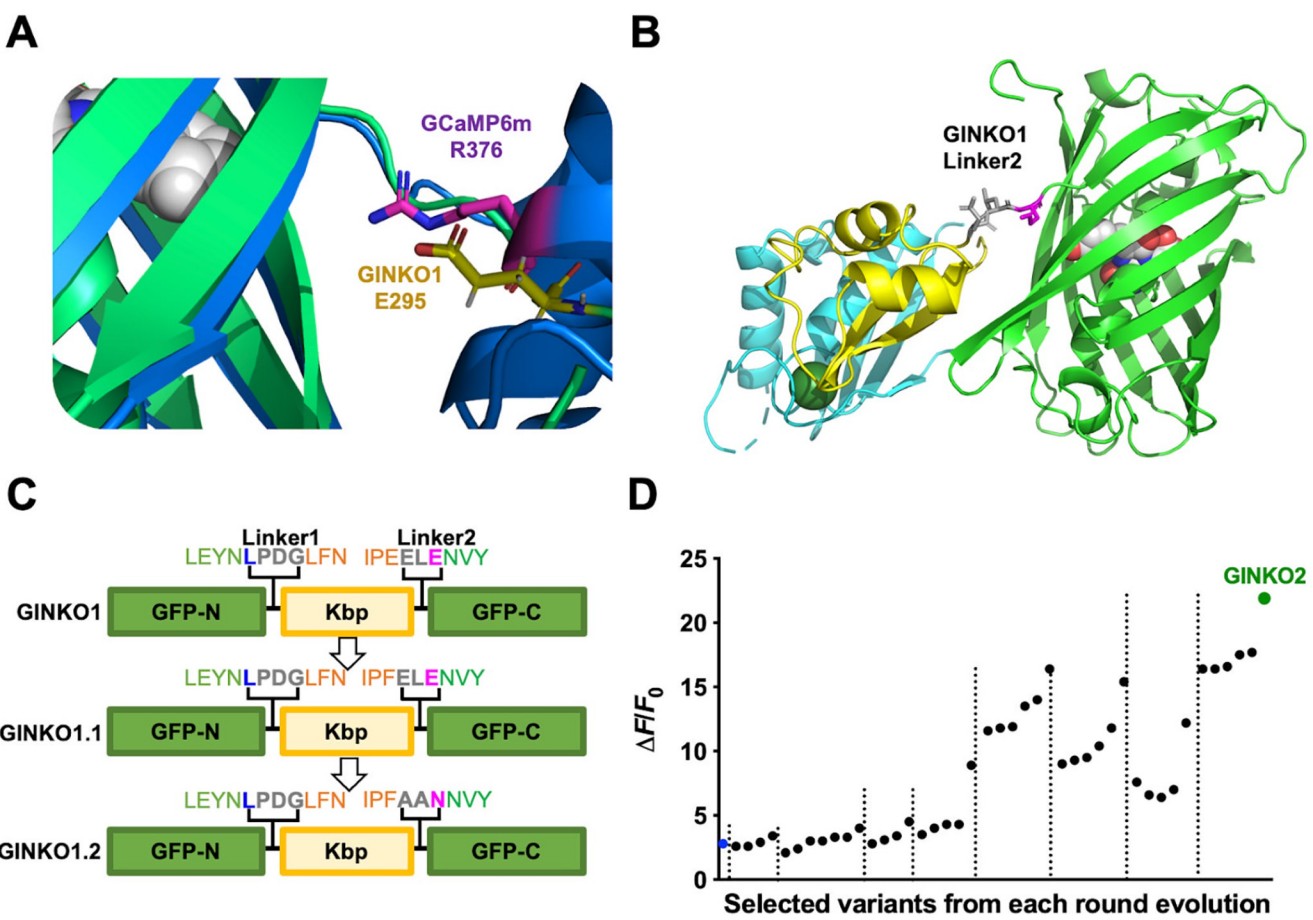

**Fig 2. Structure-guided optimization and directed evolution on GINKO1.** (A) Crystal structure alignment of GINKO1 and GCaMP6m. Alignment of R376 (magenta sticks) of GCaMP6m (PDB: 3WLC) to E295 (yellow sticks) of GINKO1. GINKO1 is represented by green ribbons, and GCaMP6m is represented by blue ribbons. Both residues point toward the chromophore of the EGFP (sphere representation). (B) GINKO1 Linker2 region, highlighted using stick representation. (C) Structure-guided optimization of GINKO. Amino acid sequences of linker regions of GINKO1, GINKO1.1, and GINKO1.2 are labeled. Green-colored residues are on GFP, orange-colored residues are on Kbp, gray colored residues are on linkers, and blue-colored L and magenta-colored E (N in GINKO1.2) are the positions of "gatepost" residues that define the optimal insertion points in EGFP [22]. (D) Selected variants in the directed evolution of GINKO. Each dot represents a variant that was selected for its improved $\Delta F/F_0$ in the lysate screening. GINKO1.2 is represented by the blue solid circle. The final variant GINKO2 is highlighted in green. The dotted lines separate libraries. The underlying data for Fig 2D can be found in S1 Data. EGFP, enhanced green fluorescent protein; Kbp, K$^+$-binding protein.

GINKO2 emission exhibits a 15× intensiometric increase at its peak of 515 nm (**Fig 3A**). GINKO2 also exhibits a ratiometric change in excitation spectrum ($\Delta R/R_0 = 20.0 \pm 0.4$, where $R$ represents the excitation ratio of 500 nm/400 nm), enabling ratiometric detection of K$^+$ concentration (**Fig 3B**). The ratiometric excitation is also observed in 2-photon (2P) characterization with the maximum fold change of 8.1 at the 2P excitation wavelength of 960 nm (**Fig 3C**). GINKO2 has a 1-photon brightness of 16 mM$^{-1}$ cm$^{-1}$ in the K$^+$-bound state, a 1.8× improvement over GINKO1 (8.6 mM$^{-1}$ cm$^{-1}$) (S3 **Table**). The 2P brightness of GINKO2 is $4.1 \pm 0.6$ GM in the K$^+$-bound state (S4 **Table**). The affinity ($K_d$) of purified GINKO2 for K$^+$ is 15.3 mM. While GINKO1 shows substantial sodium (Na$^+$)-dependent fluorescence response at concentrations below 150 mM, complicating applications where Na$^+$ is abundant [13], GINKO2 is not responsive to Na$^+$ at concentrations up to 150 mM, thus showing an improved specificity (**Fig 3D**). As the affinity for K$^+$ of GINKO2 (15.3 mM) is substantially lower than that of GINKO1 (0.42 mM) (S3 **Table**), the affinity for Na$^+$ may have also decreased

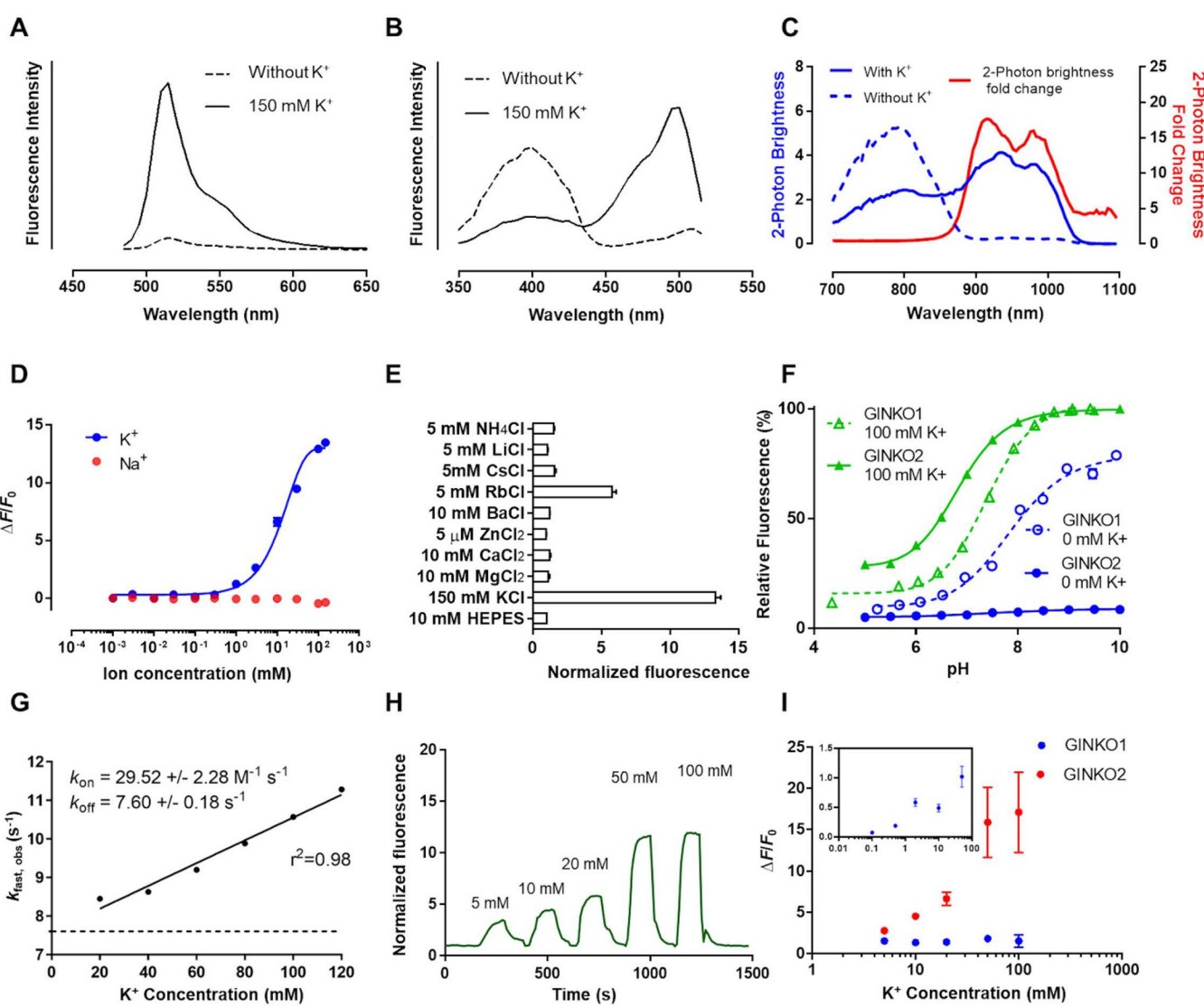

**Fig 3. GINKO2 characterization exhibited better sensitivity and selectivity.** (A) Emission spectra for GINKO2. (B) Excitation spectra for GINKO2. (C) Two-photon (2P) spectra of GINKO2. The 2P excitation spectra of GINKO2 in $K^+$-free (dash line) and $K^+$-saturated (solid line) states are colored in blue. The 2P $K^+$-dependent response of GINKO2 versus 2P excitation wavelength is colored in red. (D) $K^+$ and $Na^+$ titration of GINKO2. (E) Ion specificity of GINKO2 ($n = 3$). The concentrations of cations used were above their physiological concentrations. (F) pH titrations of GINKO1 and GINKO2. For each variant, fluorescence intensity is normalized to the maximum fluorescence. Green triangles and lines represent the presence of 100 mM $K^+$; blue circles and lines represent the absence of $K^+$. Solid symbols and lines represent GINKO2; empty symbols and dotted lines represent GINKO1. (G) Kinetics of GINKO2 ($n = 3$). (H) Representative in situ $K^+$ titration with digitonin-permeabilized HeLa cells. (I) GINKO1 ($n = 6$) and GINKO2 ($n = 10$) response curves based on in situ titration in HeLa cells. GINKO1 response curve from 0.1 to 50 mM $K^+$ is shown in the inset ($n = 17$). The underlying data can be found in S1 Data.

proportionally. Since the $K_d$ value for $Na^+$ of GINKO1 is 153 mM [13], a proportionally increased $Na^+$ $K_d$ in GINKO2 would be well outside of the physiologically relevant range of $Na^+$ concentrations. GINKO2 responds to $Rb^+$, which has an ionic radius similar to that of $K^+$, but does not respond to $Zn^{2+}$, $Mg^{2+}$, $Ca^{2+}$, $Ba^{2+}$, $Cs^+$, $Li^+$, or $NH_4^+$ at physiologically relevant concentrations (**Figs 3E, S5, and S6**). $Rb^+$ is unlikely to interfere with GINKO2 biosensing (**S5 Fig**) due to its low abundance in living organisms [23], except when used as a substitute for $K^+$ in certain experimental conditions. In addition, the $K^+$-sensing ability of GINKO2 is not affected by the presence of $Na^+$, $Zn^{2+}$, $Mg^{2+}$, $Ca^{2+}$, $Ba^{2+}$, $Cs^+$, $Li^+$, or $NH_4^+$, according to the

ion competition assay (S6 Fig). GINKO2 ($pK_a$ = 6.8 in the $K^+$-bound state) inherited the pH sensitivity of GINKO1 ($pK_a$ = 7.4 in the $K^+$-bound state) (Fig 3F). Accordingly, GINKO2 fluorescence is highly sensitive to physiologically relevant changes in pH, necessitating careful consideration of possible changes in pH during imaging applications. Kinetic measurements revealed a $k_{on}$ of 29.5 ± 2.3 $M^{-1}$ $s^{-1}$ and a $k_{off}$ of 7.6 ± 0.2 $s^{-1}$ (Fig 3G). In permeabilized HeLa cells, GINKO2 showed a $\Delta F/F_0$ of 17 when titrated with 5 to 100 mM $K^+$, which is a substantially larger change than that of GINKO1 ($\Delta F/F_0$ = 1.5) (Fig 3H and 3I). Overall, GINKO2 displays superior sensitivity and specificity over GINKO1.

## Monitoring intracellular $K^+$ concentration in bacteria with GINKO2

To determine whether GINKO2 could be used to monitor intracellular $K^+$ in bacteria, we attempted to use it in *E. coli* to monitor the decreasing intracellular $K^+$ concentration that can be induced by growth in a low-$K^+$ medium (Fig 4A). Real-time detection of intracellular $K^+$ concentration dynamics could allow the relationship between extracellular low-$K^+$ availability, intracellular $K^+$ concentration, and bacterial growth rate, to be established. The excitation

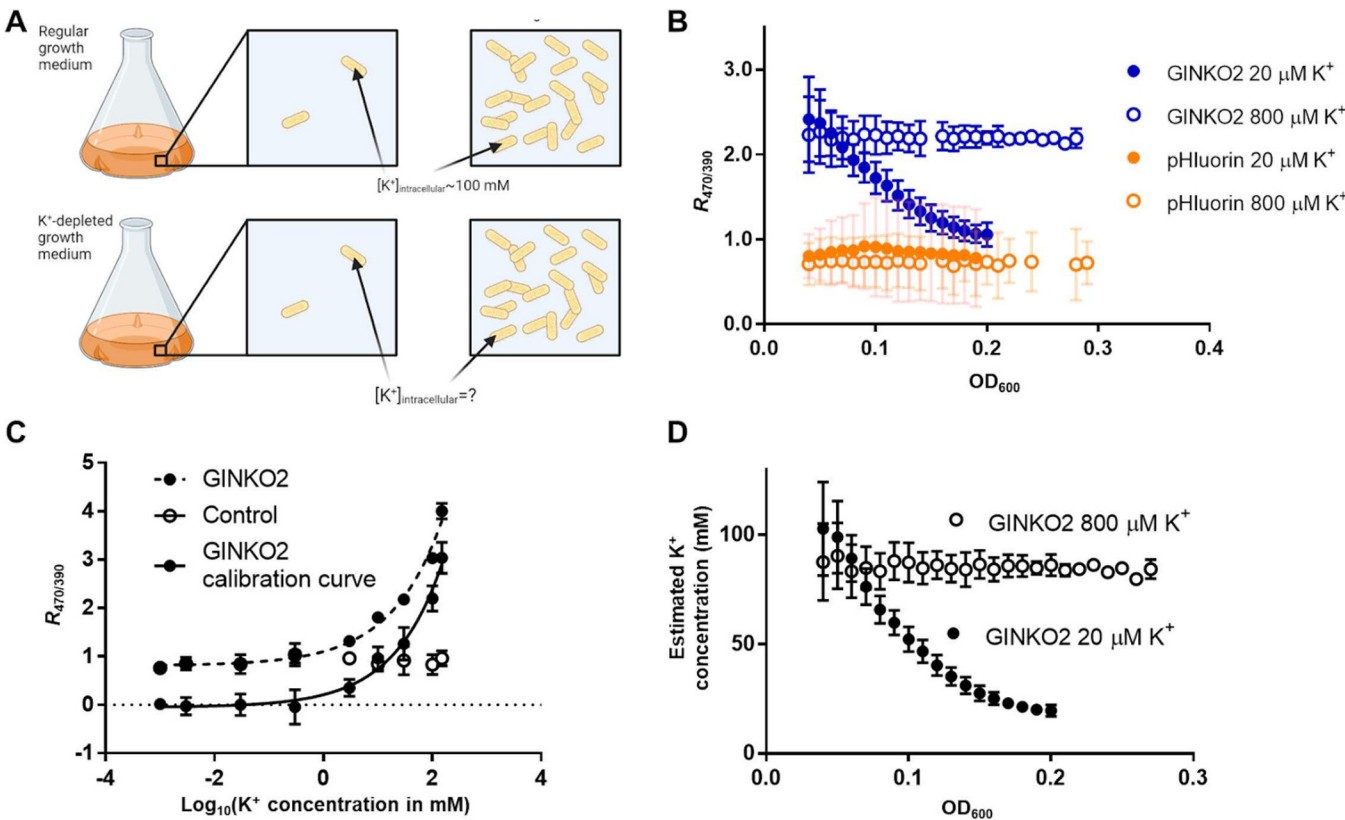

**Fig 4. Monitoring intracellular $K^+$ concentrations with GINKO2 in *E. coli* grown in $K^+$-depleted media.** (A) *E. coli* are capable of accumulating $K^+$ to a higher concentration than the environment. The free intracellular $K^+$ concentration is around 100 mM when cells are cultured with sufficient $K^+$ in the environment such as in LB. In this work, we aimed to investigate the intracellular $K^+$ concentrations of *E. coli* growing in $K^+$-depleted media. (B) Excitation ratio ($R_{470/390}$) of GINKO2 in *E. coli* cells grown in $K^+$-deficient media. Optical density at 600 nm ($OD_{600}$) reflects cell density during the growth. Two low $K^+$ concentrations (open circle: 800 μM, solid circle: 20 μM) were used for the experiment: only the medium supplemented with 20 μM $K^+$ induced detectable $K^+$ decrease during the growth. $n$ = 6–8 for *E. coli* expressing GINKO2 in 20 μM $K^+$; $n$ = 3–8 for *E. coli* expressing GINKO2 in 800 μM $K^+$; $n$ = 3 for *E. coli* expressing pHluorin in 20 μM $K^+$; $n$ = 3–6 for *E. coli* expressing pHluorin in 800 μM $K^+$. (C) A $K^+$ titration calibration curve was obtained with *E. coli* cells pretreated with 30 nM valinomycin for 5 min. The GINKO2-expressing cells (solid circle and dashed line) and nonexpressing cells (control, empty circle) were both titrated with $K^+$ at $OD_{600}$ approximately 0.1. The calibration curve (solid circle and continuous line) was obtained by subtracting the fluorescence readings of control from those of GINKO2-expressing cells. (D) $K^+$ concentrations in (B) were estimated based on the calibration curve in (C). Fig 4A was created with BioRender.com. The underlying data for Fig 4B-4D can be found in S1 Data.

ratiometric change of GINKO2 presents a unique solution to monitor K$^+$ concentration changes in proliferating *E. coli*, in which intensity-based measurements are impeded by the increasing biosensor expression level during cell growth. GINKO2-expressing *E. coli* grown in a medium with 20 μM K$^+$ exhibited a 58% decrease in excitation ratio $R_{470/390}$ (**Fig 4B**), corresponding to an estimated decrease in intracellular K$^+$ from 103 ± 21 mM to 20 ± 3 mM based on a calibration in *E. coli* (**Fig 4C and 4D**). In contrast, cells grown in a medium with 800 μM K$^+$ showed unchanged intracellular K$^+$ concentration at around 80 mM during the same growth period (**Fig 4D**). An excitation ratiometric pH biosensor pHluorin [24] was used to confirm that the intracellular pH remained stable. This application of GINKO2 demonstrated its practicality for real-time monitoring of intracellular K$^+$ in *E. coli*.

## In vivo imaging of intracellular K$^+$ dynamics in plants with GINKO2

To evaluate the utility of GINKO2 in vivo in plants, we attempted to use GINKO2 to monitor intracellular K$^+$ concentration changes in *Arabidopsis thaliana* under stress conditions. K$^+$ is an essential nutrient for plants and regulates root growth, drought resistance, and salt tolerance [25,26]. Despite the importance of K$^+$, its detailed spatiotemporal dynamics remain elusive in plants, largely due to the lack of high-performance imaging probes.

*A. thaliana* stably transformed with GINKO2 expressed under the control of the g10-90 constitutive promoter exhibited homogeneous fluorescence in leaf epidermis, hypocotyls, primary root tips, and primary mature roots (**Fig 5A**). GINKO2 expression did not affect root elongation (**S8 Fig**) nor the overall plant development. GINKO2 fluorescence was visible in the cytoplasm but absent in vacuoles. Vacuoles are K$^+$ reservoirs with concentrations as high as 200 mM. This significant store of vacuolar K$^+$ is available to be released into the cytoplasm for the regulation of the cytoplasmic K$^+$ concentration [27]. Due to the low vacuolar pH (pH = 5.0 to 5.5) [28], GINKO2 fluorescence would be quenched if it was targeted to vacuoles. Therefore, even if it was localized to the vacuole, GINKO2 is likely to be unsuitable for reporting vacuolar K$^+$ concentration changes. When the seedlings were transferred from the plant standard growing medium (½MS medium) with 10 mM K$^+$, to K$^+$ gradient buffers (0.1, 1, 10, and 20 mM) for 2.5 d, cytosolic GINKO2 fluorescence reported no significant differences in

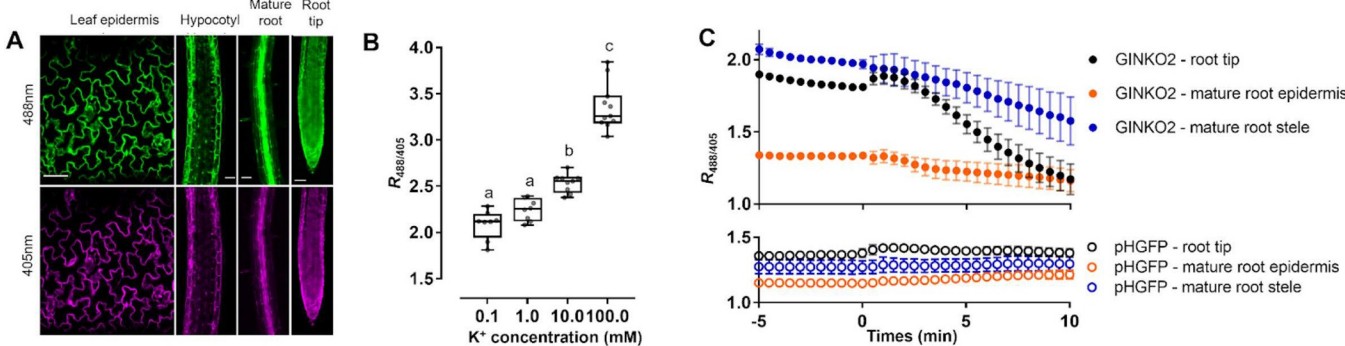

**Fig 5. Monitoring K$^+$ efflux in *Arabidopsis thaliana* with GINKO2 during salt stress.** (A) Expression and characterization of GINKO2 in *A. thaliana*. Representative fluorescence images of g10-90::GINKO2 expressing tissues excited at 405 nm and 488 nm. Scale bar = 50 μm. (B) Effect of increasing concentrations of KCl and 2 μM valinomycin on g10-90::GINKO2 $R_{488/405}$ after 6 h of K$^+$ depletion with a 0-mM KCl and 2-μM valinomycin pretreatment. $n$ = 16–21 individual roots. Letters indicate the significantly different statistical groups with $P < 0.05$ minimum. Statistical analysis was conducted with a nonparametric multiple comparison. (C) Effect of 100 mM NaCl on g10-90::GINKO2 $R_{488/405}$ (top panel) in root tips, mature root stele, and epidermis with K$^+$ depleted for 6 h. Treatment was applied at time 0. $n$ = 14 (root tip), 8 (mature root stele and epidermis) individual seedlings. pHGFP expressing roots (bottom panel) were used as controls. $n$ = 9 individual seedlings for root tips, mature root stele, and epidermis. The underlying data for Fig 5B and 5C can be found in S1 Data.

$R_{488/405}$ across the concentration range (**S9A Fig**), suggesting that the vacuolar pools of $K^+$, invisible to GINKO2, might buffer the low $K^+$ in the treatments.

It has been previously reported that during low $K^+$ treatment, the vacuolar pool of $K^+$ gradually decreases to sustain the cytosolic pool, and only when the vacuolar pool is severely diminished does the cytosolic $K^+$ concentration start to decline [27]. Therefore, we thought to deplete the vacuolar $K^+$ before imaging to reduce its buffering effect by transferring the seedlings onto a medium containing 0 mM $K^+$ and the $K^+$-specific ionophore valinomycin (2 μM). This predepletion of $K^+$ enabled the direct manipulation of the cytosolic $K^+$ concentration using media of different $K^+$ concentrations, allowing GINKO2 to display its full sensing capacity. In permeabilized and $K^+$-depleted seedlings, we observed a significant decrease of the GINKO2 $R_{488/405}$, indicating a lowered cytoplasmic $K^+$ concentration (**S9B Fig**). GINKO2 excitation ratio $R_{488/405}$ correlated well with the medium $K^+$ concentrations in the physiological range of 1 to 100 mM (**Fig 5B**).

We next imaged $K^+$ dynamics in roots under salt (NaCl) stress. The $Na^+$ influx to the roots triggers a $K^+$ efflux to counterbalance the membrane depolarization [29]. NaCl treatment without predepletion of $K^+$ produced an initial increase in the cytoplasmic $K^+$ concentration followed by a decrease after 10 min (**S9C Fig**). This, again, could be attributed to the vacuoles exporting $K^+$ into the cytoplasm. With $K^+$ predepletion and a treatment of 100 mM NaCl, GINKO2 reported the $K^+$ efflux with substantial decreases in $R_{488/405}$ in root tips (35%), mature root stele (19%), and mature root epidermis (13%) (**Fig 5C,** top panel, **S10 Fig** and **S1** and **S2 Movies**).

While cytosolic pH of plant cells is known to be tightly regulated and well maintained [30], even under an induced salt stress [31], we investigated the possibility that pH changes could be responsible for the observed changes in GINKO2 fluorescence. We used the ratiometric pHGFP, a pH sensor modified from ratiometric pHluorin for plant expression, which exhibits an increase in $R_{488/405}$ with a decrease in pH [32,33]. Ratiometric measurement of pHGFP fluorescence suggested intracellular pH remained relatively stable after the NaCl treatment (**Fig 5C,** lower panel). Specifically, in root tips, pH is transiently lowered (3% increase in pHGFP $R_{488/405}$) upon the addition of NaCl but quickly returned to the baseline level. In mature root stele, the pH remained unchanged throughout the experiment. These pH control experiments suggested that the observed decline in GINKO2 ratio under salt stress (**Fig 5C**) resulted from a change of $K^+$ concentration rather than pH. In contrast, in the mature root epidermis, pHGFP reported an overall 5% $R_{488/405}$ increase, indicating a slight pH decrease. Accordingly, we were unable to conclude that the observed $R_{488/405}$ change (13%) of GINKO2 in the epidermis was solely caused by a decrease in the $K^+$ concentration.

Taken together, these results demonstrated that GINKO2 is capable of reporting cytoplasmic $K^+$ dynamics in vivo in the roots of *A. thaliana* with great sensitivity and have provided insight into the complexity of $K^+$ regulation in plants. With appropriate protocols and controls, GINKO2 represents a substantial step forward for the study of $K^+$ homeostasis in plants with the potential to be applied to a variety of experimental paradigms, including detection and characterization of mutant phenotypes (e.g., mutations in $K^+$ transporters), and characterization of changes in $K^+$ dynamics under stress conditions.

## In vivo imaging of extracellular $K^+$ changes in mice with GINKO2

To further explore GINKO2 applications, we tested whether GINKO2 is capable of reporting extracellular $K^+$ changes in vivo during cortical spreading depolarization (CSD) in the mouse brain. CSD is a propagating, self-regenerating wave of neuronal depolarization moving through the cortex and is associated with severe brain dysfunctions such as migraine aura and seizures [34]. On the molecular level, CSD is accompanied by propagating waves of increased

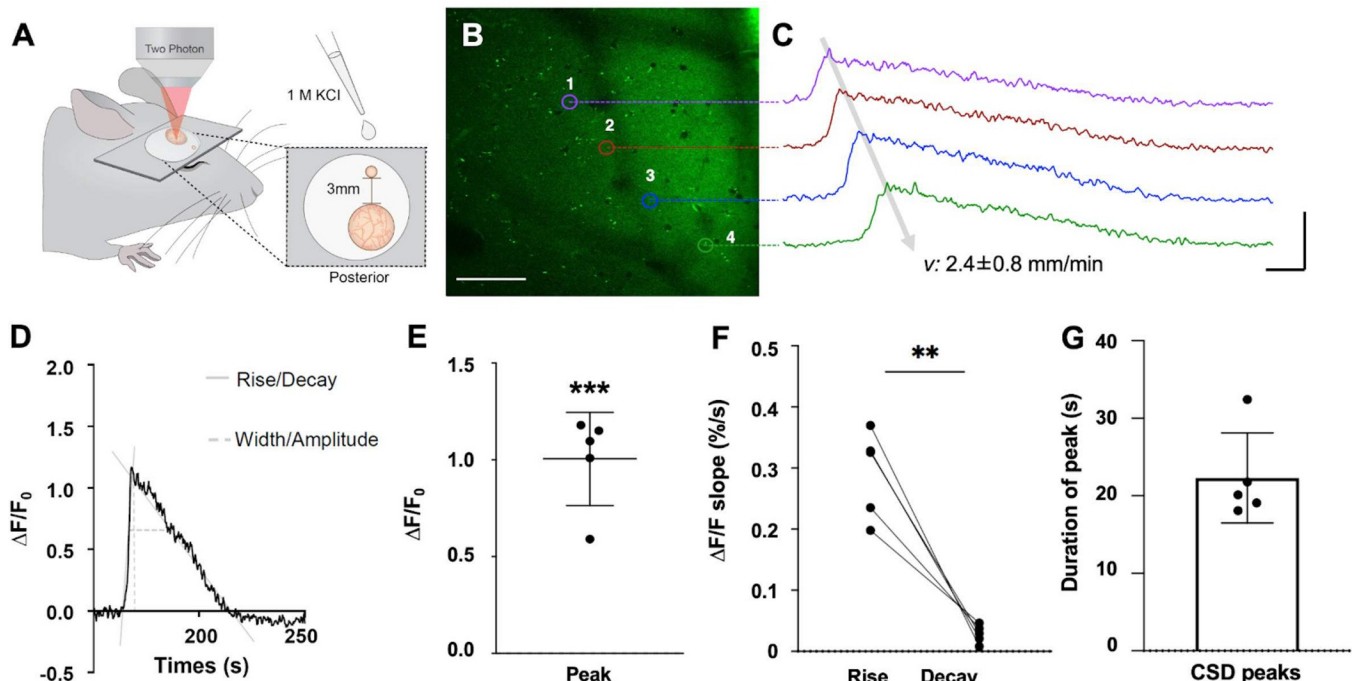

**Fig 6. Monitoring the CSD-induced elevation of extracellular K$^+$ concentrations in mice.** (A) Experimental setup of 2P microscopy in anesthetized mice. CSD was induced using 1 M KCl applied to a separate frontal craniotomy (small circle) of the imaging window (large circle) at a distance of 3 mm. Exogenously expressed GINKO2 protein was purified and externally applied to the imaging site by pipetting. (B) Averaged image of GINKO2 in the somatosensory cortex (−70 μm) obtained using 2P microscopy. *E. coli* expressed GINKO2 was applied externally by bath application 1 h before imaging. The image depicts the ROIs corresponding to traces in (C). Scale bar: 100 μm. (C) Example of traces from ROIs in the same animal, depicting the first CSD wave. x-axis: 5 s, y-axis: 100% ΔF/F$_0$, mean ± SD. (D) Example of a CSD wave showing decay, rise, width, and amplitude. (E) Comparison between ΔF/F of baseline before each CSD and at peak. $N = 2$, $n = 5$, paired $t$ test, ***$p = 0.0007$. (F) Calculated slope coefficient using simple linear regression of the rise and the decay of CSD waves. $N = 2$, $n = 5$, paired $t$ test, **$p = 0.0024$. (G) Average CSD wave duration $N = 2$, $n = 5$, mean ± SD. The underlying data for Fig 6D-G can be found in S1 Data. CSD, cortical spreading depolarization; ROI, region of interest; 2P, 2-photon.

extracellular K$^+$ from a baseline of 2.5 to 5 mM to a peak concentration of 30 to 80 mM [35]. As previously reported for Kbp-based K$^+$ biosensor GEPII [36], we have been unable to express and display functional GINKO2 on the extracellular membrane for reasons that remain unclear to us. To circumvent this limitation, we turned to the exogenous application of bacterially expressed GINKO2 as an alternative method to evaluate extracellular K$^+$ concentration dynamics during CSD. Purified GINKO2 protein (6.55 mg/mL in artificial cerebrospinal fluid (aCSF)) was exogenously applied to the extracellular space of deeply anesthetized mice above the somatosensory cortex (**Fig 6A**). To experimentally elicit CSD, we applied 1 M KCl to a separate frontal craniotomy [35] (**Fig 6A**), after which multiple waves of GINKO2 fluorescence intensity increase were observed, propagating at a velocity of 2.4 ± 0.8 mm/min (**Figs 6B, 6C, 6D and S11A and S3 Movie**). The fluorescence intensity increased by 1.0 ± 0.2× (**Fig 6E**), with a fast rise at 0.29 ± 0.07% s$^{-1}$ and a significantly slower decay at 0.03 ± 0.01% s$^{-1}$ (**Fig 6F**). The duration of the waves (width at half maximum) was 22 ± 6 s (**Fig 6G**). The fluorescence increases observed with GINKO2 during CSD (**Fig 6**) correspond well to descriptions of the extracellular K$^+$ concentration dynamics previously reported during CSD [35]. A control experiment using EGFP (2.13 mg/mL in aCSF) was performed to evaluate pH changes under the same treatment (**S11 Fig**). A 30% fluorescence decrease under the same treatment indicated a possible decrease in pH based on the pH profile of EGFP [37]. A decline in pH, suggested by either the EGFP control or previous reported pH dynamics during CSD (short increase in pH for approximately 5 s, followed by a decrease in pH [38]), should have resulted

in a GINKO2 fluorescence change in the opposite direction of the one we observed. This strongly supported that the observed elevation of GINKO2 fluorescence resulted from a substantial extracellular $K^+$ concentration increase during CSD. Overall, these results suggest that GINKO2 is an effective tool for reporting extracellular $K^+$ concentration changes in vivo in the mouse brain during CSD.

## In vivo imaging of $K^+$ dynamics in *Drosophila* neurons and glial cells with GINKO2

In an attempt to visualize potential $K^+$ changes in vivo in *Drosophila*, we fused GINKO2 with a red fluorescent pH biosensor, pHuji [39], to monitor both $K^+$ and pH concurrently. We first characterized pHuji-GINKO2 fusion protein in vitro. Decreasing pH reduces the green fluorescence of GINKO2 but does not change the affinity for $K^+$ (**S12 Fig**). The red fluorescence of pHuji is not sensitive to the $K^+$ concentration. We then produced transgenic flies expressing pHuji-GINKO2 under control of the Gal4-UAS system, either in neurons (elav-Gal4) or in glia (repo-Gal4). Fly brains were stimulated either by rapidly elevating the extracellular $K^+$ concentration or electrically with a glass electrode. In neurons, stimuli led to a decline in GINKO2 fluorescence, while in glia, the same stimuli led to an increase in GINKO2 fluorescence (**Fig 7**). However, these stimuli also led to similar changes in pHuji fluorescence, indicating substantial pH changes (**Fig 7**). It is expected that stimulated neuronal activities would likely lead to a $K^+$ efflux, as previously observed by others in several different preparations [40]. However, due to the susceptibility of GINKO2 to pH interference, the GINKO2 fluorescence changes observed in this particular set of experiments cannot be conclusively interpreted as $K^+$ changes in the stimulated neurons or glial cells.

## Considerations of pH changes for $K^+$ measurement with GINKO2

Fluorescent protein-based biosensors are often pH-sensitive [41,42], which complicates the interpretation of results obtained under conditions in which pH changes do or could occur. The p$K_a$ of GINKO2 (6.8) is very close to physiological pH (**Fig 3F**), and changes in pH could induce fluorescence intensity changes that could be misinterpreted as being induced by $K^+$ concentration changes. Accordingly, the pH sensitivity of GINKO2 had to be taken into careful consideration when we applied GINKO2 in bacteria, plants, and mice (**Figs 4, 5, 6 and S11**). In the *E. coli* growth experiment, control experiments with the pH indicator pHluorin revealed a near-constant intracellular pH under the experimental conditions (**Fig 4**). In the plant imaging experiments, pHGFP was used as a pH control, which confirmed that the fluorescence change observed in root tips and mature root stele resulted from $K^+$ concentration change (**Fig 5**). In the mice CSD experiment, an EGFP control suggested that a decrease in extracellular pH accompanied CSD (**S11 Fig**). A decrease in pH would be expected to result in decreased fluorescence of GINKO2. Accordingly, the observed increase in GINKO2 fluorescence is fully consistent with, and best explained by, a CSD-dependent increase in extracellular $K^+$ concentration (**Fig 6**). In contrast, our attempts to visualize $K^+$ change in vivo in *Drosophila* illustrated why caution must be exercised when using GINKO2 due to its pH sensitivity. In control experiments, the fluorescence changes of the pH indicator pHuji were in the same direction with similar magnitude as those of GINKO2 under the experimental conditions. Thus, we could not conclusively rule out pH as being the cause of GINKO2 fluorescence response. As with practically all GFP-based biosensors, GINKO2 is well poised for applications as long as pH remains constant or results in a GINKO2 signal change that is in the opposite direction to that caused by $K^+$. As we have demonstrated, the pHuji-GINKO2 construct provides a way to monitor pH changes in the red emission channel and may be generally useful

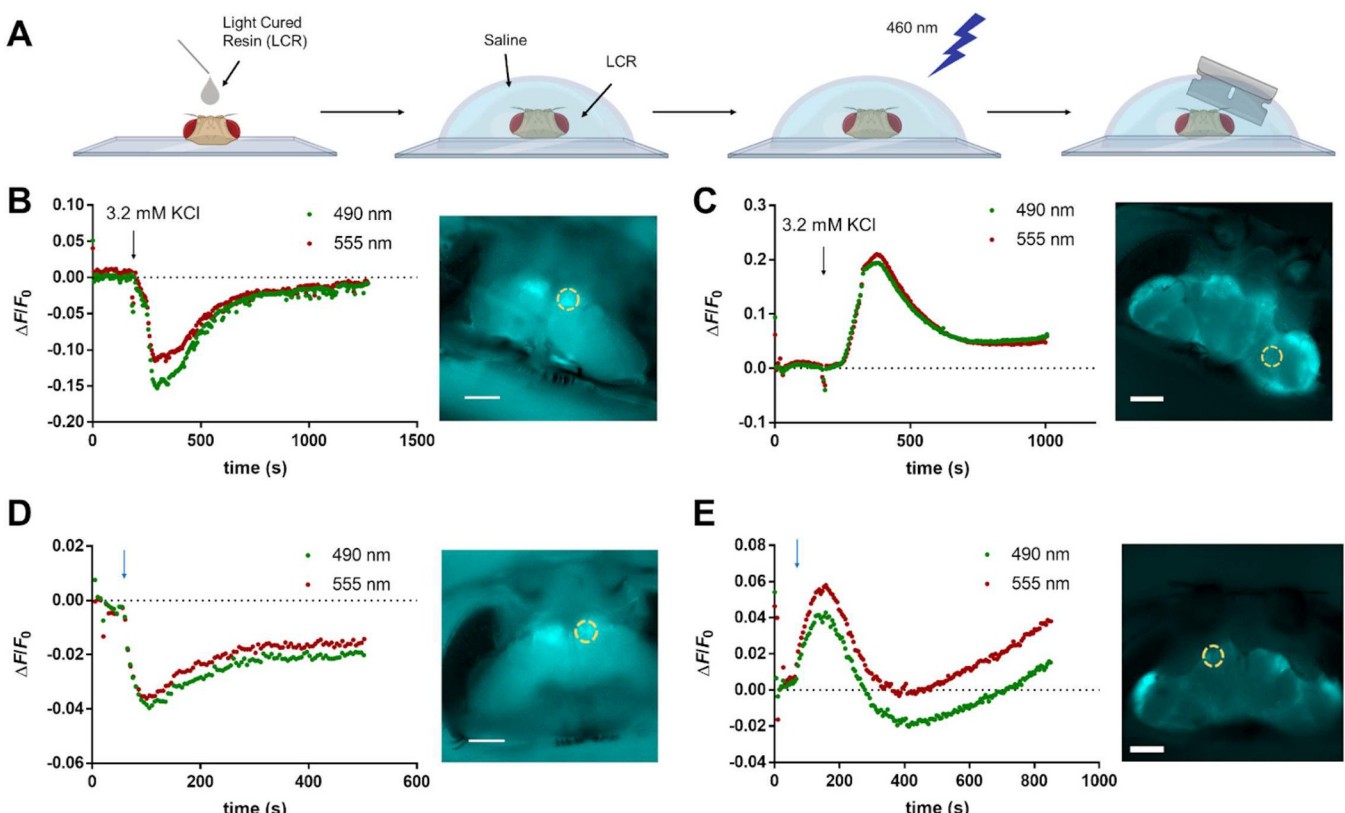

**Fig 7. pHuji-GINKO2 responses to K$^+$ or electrical stimulation in the *Drosophila* brain.** (A) Fly heads were encapsulated in a photopolymerizable resin (LCR) delivered by a thin needle with the posterior side of the head on the bottom of the petri dish. The LCR-coated heads were covered by a droplet of saline and cured by blue light at 460 nm. The heads then were transversely sectioned through the joints between the second and third antennal segments [58]. Fly brains expressing pHuji-GINKO2 in (B) neurons and (C) glia were stimulated by adding KCl in the bath to a final concentration of 3.2 mM. The black arrow indicates the time at which KCl was added. Fly brains expressing pHuji-GINKO2 in (D) neurons and (E) glia were stimulated by 500 electrical impulses delivered at 50 Hz, starting at the time indicated by the blue arrow, by a glass microelectrode. The heads were oriented with the eyes at the top of the frame during image acquisition. The samples were excited by alternating between 490 nm and 555 nm, and the ROIs used to plot the graphs are indicated by dashed circles. Scale bars: 100 μm. Fig 7A was created with BioRender.com. The underlying data for Fig 7B-7E can be found in S1 Data. LCR, light cured resin; ROI, region of interest.

for avoiding misinterpretation of GINKO2 fluorescence changes. Alternatively, it has been suggested that excitation of the protonated chromophore at approximately 400 nm could be used for pH-insensitive measurement in GFP-based excitation ratiometric indicators [43]. To further address this challenge, future efforts could be directed toward developing a less pH-sensitive GINKO variant.

## Conclusion

Here, we have engineered an improved genetically-encoded green fluorescent K$^+$ biosensor GINKO2. Due to its excellent sensitivity and specificity, this new biosensor, when used with appropriate controls for pH-dependent changes, opens new avenues for in vitro and in vivo K$^+$ imaging in a variety of model organisms.

## Methods

### Protein engineering

pBAD and pcDNA plasmids containing the gene encoding GINKO1 were used as the templates for this work. Gene fragments and primers were ordered from Integrated DNA

Technology (IDT). *E. coli* DH10B (Thermo Fisher Scientific) was used for cloning and protein expression. Site-directed mutagenesis was performed with the QuikChange lightning kit (Agilent) according to the manufacturer's instructions. Random mutagenesis was introduced via error-prone PCR (EP-PCR). Briefly, EP-PCR was performed using Taq polymerase and the standard Taq buffer (New England Biolabs) with imbalanced dNTP (0.2 mM dATP, 0.2 mM dTTP, 1 mM dGTP, and 1 mM dCTP) and modifications of $MnCl_2$ and $MgCl_2$ concentrations. A final concentration of 5.5 mM $MgCl_2$ was added to the supplier's standard reaction buffer. $MnCl_2$ was added to a final concentration of 0.15 mM and 0.30 mM to generate libraries with low-frequency and high-frequency mutations. DMSO was added at 2% (v/v) to stabilize the unmatched nucleotide pairs during the amplification. PCR products were purified on 1% agarose gel, digested with *Xho*I and *Hind*III (Thermo Fisher Scientific), and ligated with a similarly digested pBAD backbone vector using T4 DNA ligase (Thermo Fisher Scientific). The transformation of electrocompetent DH10B (Thermo Fisher Scientific) was performed with the ligation products and QuikChange products. About 5,000 to 10,000 colonies were generated for each library, among which 40 to 80 colonies with bright to medium fluorescence were picked and inoculated at 37˚C overnight. Cells were pelleted down by centrifugation at >10,000 rpm for 30 s, resuspended in 200 μL of 10 mM HEPES buffer, and lysed by 4 freeze-and-thaw cycles by alternating incubations in liquid nitrogen and 42˚C water bath. The lysate was centrifuged for 5 min and 100 μL supernatant of each sample was then transferred to a 96-well plate. The fluorescence response was read by a Safire2 microplate reader (Tecan) with excitation at 465 nm. Approximately 10 mL of 1 M KCl were then added into each well, and the fluorescence measurements were repeated. The winners were selected based on the calculated fluorescence change ($\Delta F/F_0$) and validated in triplicates. $K^+$ titrations were performed on purified variants to further verify the fluorescence change ($\Delta F/F_0$) and to determine the $K_d$. The winners were selected for the next round of optimization.

## Protein expression and purification

Single colonies of *E. coli* DH10B expressing GINKO variants were picked from the agar plate and inoculated in a flask containing 200 to 500 mL of LB supplemented with 100 μg/mL ampicillin and 0.02% (w/v) L-(+)-arabinose. The cells were cultured at 200 rpm, 37˚C for 16 to 20 h. GINKO variants were purified as previously described [13]. Briefly, the cells were pelleted by centrifugation at 6,000 rpm for 10 min and lysed by sonication. The protein was purified through affinity chromatography with Ni-NTA beads. The protein-bound beads were washed with the wash buffer supplemented with 20 mM imidazole. GINKO was eluted from the beads with the elution buffer supplemented with 500 mM imidazole. The eluted protein was then buffer exchanged to 10 mM HEPES at pH 7.4 by PD-10 columns (GE Healthcare Life Sciences) following the manufacturer's instructions.

## Crystallization and structure determination

The His-tag affinity-purified GINKO1 protein was further applied on the size exclusion chromatography Superdex200 (GE Healthcare) column preequilibrated with 25 mM Tris (pH 7.5), 150 mM KCl buffer. The main fractions of monodisperse protein were concentrated to around 25 mg/mL for crystallization trials. Crystallization experiments were set up in sitting drop geometry with 0.5 μL protein sample equilibrating with 0.5 μL reservoir from screen kits (Hampton and Molecular Dimensions) at room temperature. The final diffraction quality crystals were grown in 0.1 M MES (pH 6.0), 20% PEG6000 after several rounds of crystallization optimization (S1B Fig). For data collection, the crystals were transferred to the crystal stabilization buffer supplemented with 10% to 15% PEG400 or glycerol and flash-frozen in liquid

nitrogen. X-ray diffraction data were collected at GM/CA@APS beamline 23IDB, using the raster to identify a well-diffracting region of an inhomogeneous rod-shaped crystal, and were initially processed with the beamline supplemented software package. The X-ray diffraction data were further integrated and scaled with the XDS suite [44]. The data collection details and statistics were summarized in crystallographic S1 **Table**. The GINKO1 structure was determined with a maximum-likelihood molecular replacement program implemented in the Phaser program [45], using structures of the GFP (6GEL) and the $K^+$ binding protein (5FIM) as search models [12,46]. The linker and $K^+$ density were observed after initial refinement. The missing residues manual model rebuilding and refinement were carried out with the COOT program and the PHENIX suite [47,48]. The GINKO1 structure was solved at 1.85 Å in the P1 space group with the unit cell dimension a = 46.8 Å, b = 49.3 Å, c = 83.7 Å, and α = 89.96°, β = 89.97°, γ = 80.95°. The final structure model was refined to a $R_{work}/R_{free}$ value of 0.1947/0.2252. The model contained 2 GINKO1 molecules each containing 1 $K^+$ and 892 water molecules in the asymmetric unit cell.

## In vitro characterization

The purified GINKO variants were titrated with $K^+$ and $Na^+$ to determine the fluorescence change $\Delta F/F_0$ and the affinity. The titration buffers were prepared in 10 mM HEPES at pH 7.4 supplemented with 0.001, 0.003, 0.01, 0.03, 0.1, 0.3, 1, 3, 10, 30, 100, and 150 mM KCl or NaCl. The buffers for specificity tests were prepared in 10 mM HEPES at pH 7.4. The buffers used for pH titrations were 10 mM HEPES adjusted with NaOH or HCl to pH 5.5, 6.0, 6.5, 7.0, 7.5, 8.0, 8.5, 9.0, 9.5, and 10.0 in the presence or absence of 150 mM KCl. The fluorescence measurements were performed in a Safire2 microplate reader (Tecan). The excitation wavelength was set at 460 nm for the emission scan from 485 to 650 nm, and the emission wavelength was set at 540 nm for the excitation scan from 350 to 515 nm. The extinction coefficient (EC) and quantum yield (QY) were determined to quantify the brightness of GINKO variants as described previously [13]. Briefly, GINKO variants fluorescence was measured in 10 mM HEPES at pH 7.4 either supplemented with 150 mM KCl or free of both $K^+$ and $Na^+$. To determine EC, a DU800 spectrophotometer (Beckman Coulter) was used to measure the absorbance and quantify the denatured chromophores at 446 nm after base denaturation with 0.5 M NaOH [49]. The QY was determined using GINKO1 as the standard. Fluorescence was measured with the Safire2 microplate reader (Tecan). Rapid kinetic measurements of the interaction between GINKO2 and $K^+$ were made using SX20 stopped-flow reaction analyzer (Applied Photophysics) using fluorescence detection. The dead time of the instrument was 1.1 ms. The excitation wavelength was set at 488 nm with 2 nm bandwidth and emission was collected at 520 nm through a 10-mm path. A total of 1,000 data points were collected over 3 replicates at increments of 0.01 s for 10 s. Reactions were initiated by mixing equal volumes of diluted purified GINKO2 protein in 100 mM Tris–HCl (pH 7.20) with various concentrations of KCl (20, 40, 60, 80, 100, and 120 mM) at 20°C. Approximately 100 mM Tris–HCl buffer was used as a blank.

Two-photon excitation spectra were measured as described [50]. In the spectral shape measurement, Coumarin 540A in DMSO and LDS 798 in $CHCl_3$:$CDCl_3$ (1:2) were used as standards. A combination of 770SP and 633SP filters was used to block the laser scattering. The cross-section $\sigma_{2,A}$ was measured at 940 nm and 960 nm. The measurement was performed using rhodamine 6G (Rh6G) in methanol as a reference standard (with $\sigma_2(940) = 9 \pm 1$ GM and $\sigma_2(960) = 13 \pm 2$ GM) [50]. The 2P fluorescence signals of the sample and reference solutions in the same excitation and collection conditions were measured. For $\sigma_2$ measurement, we used a combination of the 770SP and 520LP filters in the emission channel. Measurements

at both wavelengths gave similar results. To obtain the 2P excitation spectrum in units of molecular brightness $F_2(\lambda)$, we normalized the unscaled 2PE spectrum to the product of fractional concentration, $\rho_A$, fluorescence quantum yield, $\varphi_A$, and 2P absorption cross-section, $\sigma_{2,A}$(940 nm), of the anionic form, where all 3 parameters were measured independently, as described previously [41,42]. The molecular brightness of the anionic form presented in **S4 Table** corresponds to the spectral maxima, $\lambda_{max}$, for both states of the sensor.

## Mammalian cell culture and imaging

HeLa cells were cultured in Dulbecco's Modified Eagle Medium (DMEM, Gibco) supplemented with 10% fetal bovine serum (FBS, Gibco) and 200 U/mL penicillin–streptomycin (Thermo Fisher Scientific). The HeLa cells were transfected with pcDNA-GINKO variants by TurboFect transfection reagent (Thermo Fisher Scientific) as per the manufacturer's instructions. The transfected cells were first treated with 10 nM digitonin for about 15 min in the imaging buffer (1.5 mM $CaCl_2$, 1.5 mM $MgSO_4$, 1.25 mM $NaH_2PO_4$, 26 mM $NaHCO_3$ and 10 mM D-Glucose, pH = 7.4) saturated with 95% $O_2$ / 5% $CO_2$. The cells were then imaged on an upright FV1000 confocal microscope (Olympus) equipped with FluoView software (Olympus) and a 20× XLUMPlanF1 water immersion objective (NA 1.0, Olympus) with a flow rate of 10 mL/min using a peristaltic pump (Watson-Marlow). GINKO variants were excited with a 488-nm laser, and emission was collected in the channel from 500 to 520 nm. The perfusion buffers were prepared in imaging buffers with various $K^+$ concentrations (0.1, 0.5, 2, 5, 10, 20, 50, and 100 mM). N-methyl-D-glucamine (NMDG) was supplemented to keep osmotic pressure consistent. Fluorescence images were processed in Fiji. Regions of interest (ROIs) were selected manually based on areas with green fluorescence.

## $K^+$ titration in *E. coli* cells

*E. coli* DH10B expressing GINKO2 were grown in the LB medium overnight at 37°C. Pelleted cells were resuspended in a 10-mM HEPES buffer (pH 7.4) supplemented with 30 nM valinomycin and incubated for 5 min to allow cell membrane permeabilization for $K^+$. In a 96-well plate, 10 μL of the resuspended cells was added to 100 μL of the 10 mM HEPES buffers (pH 7.4) supplemented with various concentrations of KCl (0.001, 0.003, 0.03, 0.3, 3, 10, 30, 100, and 150 mM). The fluorescence measurements were performed in a Safire2 microplate reader (Tecan). Cells that were not transformed with GINKO2 plasmid were used as the control.

## *E. coli* growth in $K^+$-depleted environment

*E. coli* NCM3722 cells were grown in a minimal medium with 20 mM $NaH_2PO_4$, 60 mM $Na_2HPO_4$, 10 mM NaCl, 10 mM $NH_4Cl$, 0.5 mM $Na_2SO_4$, 0.4% arabinose and micronutrients [51]. Micronutrients include 20 μM $FeSO_4$, 500 μM $MgCl_2$, 1 μM $MnCl_2·4H_2O$, 1 μM $CoCl_2·6H_2O$, 1 μM $ZnSO_4·7H_2O$, 1 μM $H_{24}Mo_7N_6O_{24}·4H_2O$, 1 μM $NiSO_4·6H_2O$, 1 μM $CuSO_4·5H_2O$, 1 μM $SeO_2$, 1 μM $H_3BO_4$, 1 μM $CaCl_2$, and 1 μM $MgCl_2$. KCl was added at 800 μM or 20 μM. Ampicillin was added at 100 μg/mL to LB medium cultures and 20 μg/mL to minimal medium cultures. Single colonies were picked from LB agar plates and cultured in LB medium for 3 to 5 h at 37°C in a water bath shaker at 240 rpm. Cells were then diluted 1,000 times into arabinose-containing minimal medium (800 μM KCl) and grown at 37°C in a water bath shaker at 240 rpm overnight. Cells were washed once in the minimal medium supplemented with 20 or 800 μM KCl and diluted 500× into 96-well plates with 200 μL of the same medium supplemented with 0.4% arabinose in each well (20 or 800 μM KCl). The 96-well plates were incubated at 37°C in a Spark Plate reader (Tecan). Every 7 min, a loop would run with the following actions: First, the plate was shaken for 200 s in the "orbital"

mode with an amplitude of 4.5 mm at 132 rpm; then optical density (OD) was measured at 600 nm; fluorescence was measured at 2 wavelength settings: excitation at 390 nm, emission at 520 nm; and excitation at 470 nm, emission at 520 nm. OD was binned into the nearest 0.01, and 3 or more replicates were performed for each sample. Background fluorescence of nonfluorescent wild-type *E. coli* NCM3722 control was subtracted from the fluorescence of experimental samples.

## In vivo K$^+$ imaging in plants

*A. thaliana* ecotype Columbia 0 (Col0) was used as the wild type and background for the expression GINKO2. GINKO2 was cloned into the pUPD2 plasmid using the GoldenBraid cloning system [52]. GINKO2 was placed under the control of the strong constitutive g10-90 promoter [53], terminated by the Rubisco terminator, and together with the BASTA selection cassette, combined into the binary pDGB1_omega1 vector. Stable transformation of *A. thaliana* plants was achieved by the floral dip method [54]. Transformed plants were then selected by their BASTA resistance and optimal fluorescence; single-locus insertion lines were selected for further propagation until homozygous lines were established. The pHGFP expressing *A. thaliana* was obtained as previously described [33]. Seeds were surface sterilized by chlorine gas for 2 h and sown on 1% (w/v) plant agar (Duchefa) with ½ Murashige and Skoog (½MS, containing 10 mM K$^+$ and 51 μM Na$^+$, Duchefa), 1% (w/v) sucrose, adjusted to pH 5.8 with NaOH, and stratified for 2 d at 4˚C. Seedlings were grown vertically for 5 d in a growth chamber with the temperature at 23˚C by day (16 h) and 18˚C by night (8 h), 60% humidity, and the light intensity of 120 μmol photons m$^{-2}$ s$^{-1}$. For the KCl gradient experiments, treatments were applied by transferring the plants to 0.7% (m/v) agarose (VWR Life Sciences) with 1.5 mM MES buffers (Duchefa) supplemented with various concentrations of KCl and adjusted to pH 5.8 with NaOH. To deplete cellular K$^+$, seedlings were transferred to a 0-mM KCl medium containing 2 μM valinomycin (Glentham Life Sciences, 10 mM in DMSO) in 0.7% agarose with 1.5 mM MES at pH 5.8, for 6 h before imaging. For the KCl gradient experiment, seedlings were K$^+$ depleted for 30 min before imaging. For the NaCl treatment, seedlings were transferred from solid media to custom microfluidics chips [55]. Seedlings were first imaged in the control solution (0 mM NaCl in 1.5 mM MES buffer (pH 5.8)) before switching to the treatment solution (100 mM NaCl in 1.5 mM MES buffer (pH 5.8)). A constant flow of 3 ± 0.01 μL/min was maintained using a piezoelectric pressure controller (OBI1, Elveflow) coupled with microflow sensors (MFS2, Elveflow) and the dedicated Elveflow ESI software to control both recording and the flow/pressure feedback. The root elongation toxicity assay was performed by scanning Col0 and g10-90::GINKO2 seedlings grown in square plates containing 1/2MS media for 16 h every 30 min with an Epson v370 perfection scanner. Root elongation was quantified with a semiautomated workflow [55]. Microscopy imaging was performed using a vertical stage Zeiss Axio Observer 7 with Zeiss Plan-Apochromat 20×/0.8, coupled to a Yokogawa CSU-W1-T2 spinning disk unit with 50 μm pinholes and equipped with a VS401 HOM1000 excitation light homogenizer (Visitron Systems). Images were acquired using the VisiView software (Visitron Systems). GINKO2 and pHGFP were sequentially excited with 488 nm and 405 nm lasers and the emission was filtered by a 500- to 550-nm bandpass filter. Signal was detected using a PRIME-95B Back-Illuminated sCMOS Camera (1,200 × 1,200 pixels; Photometrics). For microfluidic experiments, the fluorescence was measured using the segmented line tool with a 40-pixels width. All microscopy image analyses were conducted using the software ImageJ Fiji v1.53c [56]. Statistical analyses were performed using R software. Boxplots represent the median and the first and third quartiles, and the whiskers extend to data points <1.5 interquartile range away from the first or third quartile; all data points are shown

as individual dots. We used two-sided nonparametric Tukey contrast multiple contrast tests (mctp function) with logit approximation.

## In vivo imaging of CSD in mice

All experiments performed at the University of Copenhagen were approved by the Danish National Animal Experiment Committee (2020-15-0201-00558) and were in accordance with European Union Regulations. The experiment plan was overseen by the University of Copenhagen Institutional Animal Care and Use Committee (IACUC). Male C57BL/6J wild-type mice 8 to 10 weeks old (Janvier) were used for in vivo studies. Mice were kept under a diurnal lighting condition (12 h light/12 h dark) in groups of 5 with free access to food and water. Mice were deeply anesthetized (ketamine: 100 mg/kg, xylazine: 20 mg/kg) and fixed to a stereotaxic stage with ear bars. Body temperature was maintained at 37°C with a heating pad, and eye drops were applied. A metal head plate was attached to the skull using dental acrylic cement (Fuji LUTE BC, GC Corporation, Super Bond C&B, Sun Medical). A small craniotomy for KCl application was made on the skull above the frontal cortex (AP: 1.0 mm ML: 1.2 mm). Likewise, a 3-mm diameter craniotomy for imaging was drilled above the ipsilateral somatosensory cortex (AP: −1.5 mm, ML: 2.0 mm). To prepare the window for imaging, the dura was carefully removed before sealing half the craniotomy with a thin glass coverslip (3 mm × 5 mm, thickness: 0.13 mm, Matsunami Glass) using dental cement. Two-photon imaging was performed with a B-Scope equipped with a resonant scanner (Thorlabs), a Chameleon Vision 2 laser (Coherent, wavelength 940 nm), and an Olympus objective lens (XLPlan N × 25). The filter set for the detection of the green channel was as follows: primary dichroic mirror ZT405/ 488/561/680-1100rpc (Chroma); secondary dichroic mirror FF562-Di03 (Semrock); emission filter: FF03-525/50 (Semrock). The power after the objective lens ranged between 15 mW and 30 mW. Images were acquired at a depth of 70 μm with a frame rate of 30 Hz. Immediately after surgery, deeply anesthetized mice were moved to the imaging stage, and 150 μL of GINKO2 (6.55 mg/mL in HEPES-aCSF) or 75 μL of EGFP (2.13 mg/mL in HEPES-aCSF) was applied to the craniotomy above somatosensory cortex 60 to 80 min before imaging. Anesthesia level was carefully monitored and maintained during the entire course of the experiment. Cortical spreading depolarization was induced by applying a small drop (50 to 150 μL) of 1 M KCl solution to the frontal craniotomy. After acute imaging procedure, mice were perfused with fixative for histology under deep anesthesia or euthanized by overdose (ketamine-xylazine >300 mg/kg, >30 mg/kg). Fluorescence images were processed in Fiji. ROIs were selected manually based on areas with green fluorescence. Areas with small intense elements of green fluorescence were avoided. The mean fluorescence intensity of each ROI was calculated and smoothed by a 3-point average filter in MATLAB. The example trace in **Fig 6D** was calculated and smoothed by a 5-point average filter in MATLAB. Thereafter, relative fluorescence changes ($\Delta F/F$) were calculated: $F$ was the mean intensity of the pre-CSD period, and $\Delta F$ was the difference between the signal and $F$. Velocity was calculated for the passage of signal intensity peak. Graphpad Prism was used to create figures. The data were represented as mean ± SD. The slope coefficient was calculated using simple linear regression in Prism 9. Shapiro–Wilk normality test and paired $t$ test were performed using Prism 9. N represents the number of biological replicates, and n presents the number of technical replicates.

## In vivo imaging in *Drosophila*

To generate transgenic flies expressing pHuji-GINKO2 under the control of the Gal4-UAS system, pHuji-GINKO2 was cloned into the pUAST vector [57]. The vector was injected into $w^{1118}$ embryos (BestGene), and transformant lines with insertions on each major chromosome

were selected. To drive expression in all neurons, UAS-pHuji-GINKO2 flies were crossed to $w^{1118}$ *elav-Gal4*$^{C155}$. To drive expression in glia, UAS-pHuji-GINKO2 flies were crossed to $w^{1118}$ *repo-Gal4/TM3, Sb*. The head capsules of flies were opened using the goggatomy procedure [58], where the head is rapidly encapsulated in a photopolymerizable resin and then sliced to expose the live brain. Heads were cut transversely along a line through the joints between the second and third antennal segments. All experiments were performed in saline with the following composition: 120 mM NaCl, 3 mM KCl, 1.5 mM $CaCl_2$, 4 mM $MgCl_2$, 4 mM $NaHCO_3$, 1 mM $NaH_2PO_4$, 8 mM D-trehalose, 5 mM D-glucose, and 5 mM TES (pH 7.2). The bath solution (approximately 2.5 mL) was oxygenated and stirred by directing an air-stream over the solution. Glass electrodes filled with saline were used to stimulate the brain and timing was controlled by an A.M.P.I. Master-8 (Microprobes for Life Science). Fly brains were imaged on a BX50WI upright microscope (Olympus) with an ORCA-Flash 4.0 CMOS camera (Hamamatsu). GINKO2 fluorescence was monitored at 510 nm with excitation at 402 and 490 nm, whereas pHuji was excited at 555 nm and the emission was monitored at 610 nm. Illumination was provided by a LED (CoolLED) through a Pinkel filter set (89400—ET—DAPI/FITC/TRITC/Cy5 Quad, Chroma). Images were acquired with MetaMorph software (Molecular Devices).

### Data analysis

Microsoft Excel was used for data analyses of GINKO characterizations and titrations. Graph-pad Prism was used to create figures. The data are represented as mean ± SD, except for the permeabilized HeLa titration, which is represented as mean ± SEM, while n represents the number of replicates.

### Supporting information

**S1 Table. X-ray data collection and refinement statistics.**
(DOCX)

**S2 Table. Mutations accumulated during directed evolution.**
(DOCX)

**S3 Table. Summary of GINKO1 and GINKO2 photophysical characteristics.**
(DOCX)

**S4 Table. GINKO2 two-photon characteristics.**
(DOCX)

**S1 Fig. Crystallization and X-ray crystallography of GINKO1.** (A) Image of GINKO1 crystals in TBS supplemented with 150 mM KCl. (B) The electron density map of the $K^+$ binding site of GINKO1. The density map was shown with $2F_o—F_c = 2.5 σ$.
(TIFF)

**S2 Fig. Comparison of E295 variants of GINKO1.** The bulky hydrophobic residues Y and F lead to improved $\Delta F/F_0$. E295W retained the $\Delta F/F_0$ but simultaneously increased the apparent $K_d$ substantially. E295K and E295R resulted in a smaller $\Delta F/F_0$ than template GINKO1. The underlying data can be found in S1 Data.
(TIFF)

**S3 Fig. The scheme of directed evolution of GINKO2.** Error-prone PCR was used to amplify the GINKO gene with random mutations. The PCR products were digested and ligated into a pBAD vector. After transformation, 5,000–10,000 colonies were visually inspected and around

40–80 were picked and cultured based on their brightness. Variants were preliminarily screened based on the colony brightness because high fluorescence intensity of a variant in bacterial cytosol (a high $K^+$ environment) could correlate to a high brightness in the $K^+$-bound state, which is desirable for a positive response biosensor. The cultures were then pelleted and proteins were extracted via freeze-and-thaw. The lysates were screened in a plate reader with fluorescence measurements in the absence and presence of $K^+$. In the secondary screening, variants with the largest fluorescence changes were selected for further characterization and winning variants were used as templates for the next iterative round of evolution. S3 Fig created with BioRender.com.
(TIFF)

**S4 Fig. Mutations positioned in the GINKO1 structure.** K356R (A and B) led to an improved fluorescence change ($\Delta F/F_0$) with a possible stronger electrostatic interaction. (A) K356 (green sticks) is in proximity to D148 (cyan sticks) to form electrostatic interaction. (B) R356 (green sticks) side chain is more extended than the K356 side chain, potentially providing a stronger charge attraction with a shorter distance to the D148 side chain. K102E and K259N (C and D) reduce charge repulsion between 2 lysine residues. (C) K102 (yellow sticks) and K259 (green sticks) repulse each other with the same charge. (D) E102 (yellow sticks) and N259 (green sticks) removed the charge repulsion, potentially leading to a stable interface between Kbp and EGFP. (E) Top view of the overall structure with mutation sites labeled and highlighted in sticks. (F) Side view of the overall structure with mutation sites labeled and highlighted in sticks.
(TIFF)

**S5 Fig. $Rb^+$ titration of GINKO2.** (A) The fluorescence change ($\Delta F/F_0$) of GINKO2 versus $Rb^+$ concentration. $n = 3$. (B) Excitation ratio ($R_{500/400}$) of GINKO2 versus $Rb^+$ concentration. $Rb^+$ concentration in physiology ranges from 1.7 μg/g tissue (uterus) to 11 μg/g tissue (brain and liver) [23]. This translates to a concentration range of 0.02–0.13 mM (assuming a tissue density of 1,000 g/L). Within this concentration range of $Rb^+$, GINKO2 does not exhibit fluorescence intensity or ratio change. The underlying data can be found in S1 Data.
(TIFF)

**S6 Fig. Ion competition assay of GINKO2.** To examine whether other cations in the environment could affect the $K^+$ response of GINKO2, a specificity test was performed in both 0 and 150 mM $K^+$ buffer ($n = 3$). Without $K^+$, only $Rb^+$ was able to induce a fluorescence change. In presence of 150 mM $K^+$, the fluorescence change induced by $K^+$ was not affected by any other cations. The underlying data can be found in S1 Data.
(TIFF)

**S7 Fig. Effect of growth medium $K^+$ concentration on *E. coli* growth rate.** Growth curves of *E. coli* in 20 μM and 800 μM $K^+$ medium. These 2 growth curves indicated that cells grown in the medium supplemented with 20 μM $K^+$ experienced a slower rate of growth, likely due to the limited availability of $K^+$. $n \geq 3$. The underlying data can be found in S1 Data.
(TIFF)

**S8 Fig. Root elongation of wild-type (WT) and GINKO2-expressing A. thaliana.** Comparison of root elongation between WT Columbia 0 *A. thaliana* ecotype to Columbia 0 seedlings expressing g10-90::GINKO2. $n = 18$ individual seedlings for the Col0 WT group, $n = 19$ individual seedlings for the g10-90::GINKO2 group. Letters indicate the significantly different statistical groups with $P < 0.05$ minimum. Statistical analysis was conducted with a

nonparametric multiple comparison. The underlying data can be found in S1 Data.
(TIFF)

**S9 Fig. GINKO2 responses in plants under various conditions.** (A) Effect of increasing concentrations of KCl on g10-90::GINKO2 $R_{488/405}$ without $K^+$ depletion pretreatment. $n \geq 10$ individual seedlings. Letters indicate the significantly different statistical groups with $P < 0.05$ minimum. Statistical analysis was conducted with nonparametric multiple comparisons. (B) Effect of a 6-h $K^+$ depletion treatment with 0 mM KCl and 2 μM valinomycin on g10-90::GINKO2 $R_{488/405}$. 1/2MS: Murashige and Skoog medium half strength. $n \geq 20$ individual seedlings. $P < 0.01$. (C) Effect of 100 mM NaCl on g10-90::GINKO2 $R_{488/405}$ in root tip without prior $K^+$ depletion. The treatment was applied at time 0. $n = 9$ individual seedlings. The underlying data can be found in S1 Data.
(TIFF)

**S10 Fig. Kymographs of the root tip and the mature root responding to salt stress.** Representative timelapse of g10-90::GINKO2 fluorescences after the treatment with 100 mM NaCl at time 0 in the root tip and the mature root depleted with a 0-mM KCl and 2-μM valinomycin pretreatment. The location of the selection is indicated in red above the pictures. Both channels are represented as a composite image. ep: epidermis, cor: cortex, end: endodermis. The root tip shrank upon the NaCl application due to the osmotic pressure change.
(TIFF)

**S11 Fig. Fluorescence change of GINKO2 and EGFP during CSD in mice.** (A) Image series of changes of GINKO2 fluorescence during CSD. The images were extracted from S3 Movie with a 2-s interval between them. (B) A representative fluorescence trace of an ROI (red squared region) displayed an approximately 30% dip in fluorescence intensity of EGFP. Images of the brain at 4 different times are shown in the lower panel.
(TIFF)

**S12 Fig. pH titration of pHuji-GINKO2 at various concentrations of $K^+$.** The titration curves (A and B) of pHuji-GINKO2 resemble those of GINKO2. (A) The excitation ratio $R_{490/390}$ of GINKO2 versus pH. (B) The maximum emission intensity of GINKO2 versus pH. (C) The maximum emission intensity of pHuji versus pH. The underlying data can be found in S1 Data.
(TIFF)

**S1 Movie. Salt stress-induced $K^+$ decrease in the root tip of $K^+$-depleted *A. thaliana*.** (MOV)

**S2 Movie. Salt stress-induced $K^+$ decrease in the mature root of $K^+$-depleted *A. thaliana*.** (MOV)

**S3 Movie. Real-time monitoring of CSD waves with GINKO2 in mice.** (MOV)

**S1 Data. Summary of numerical values used for data plots and statistical analysis.** (XLSX)

## Acknowledgments

SYW and YS thank the University of Alberta Molecular Biology Services Unit for technical assistance. NBCS and MF thank Eva Medvecká for technical support. ARK and DFE thank Walter Boron for helpful suggestions for the *Drosophila* experiments.

## Author Contributions

**Conceptualization:** Robert E. Campbell, Yi Shen.

**Data curation:** Sheng-Yi Wu, Yurong Wen, Nelson B. C. Serre, Cathrine Charlotte Heiede Laursen, Andrea Grostøl Dietz, Brian R. Taylor, Mikhail Drobizhev, Rosana S. Molina, Abhi Aggarwal, Vladimir Rancic, Michael Becker, Daniel F. Eberl, Alan R. Kay.

**Formal analysis:** Sheng-Yi Wu, Yurong Wen, Nelson B. C. Serre, Andrea Grostøl Dietz, Brian R. Taylor, Mikhail Drobizhev, Abhi Aggarwal, Daniel F. Eberl, Alan R. Kay, Yi Shen.

**Funding acquisition:** Mikhail Drobizhev, Klaus Ballanyi, Hajime Hirase, Maiken Nedergaard, Matyáš Fendrych, M. Joanne Lemieux, Daniel F. Eberl, Alan R. Kay, Robert E. Campbell.

**Investigation:** Sheng-Yi Wu, Robert E. Campbell, Yi Shen.

**Methodology:** Sheng-Yi Wu, Brian R. Taylor.

**Project administration:** Robert E. Campbell, Yi Shen.

**Supervision:** Klaus Ballanyi, Kaspar Podgorski, Hajime Hirase, Maiken Nedergaard, Matyáš Fendrych, M. Joanne Lemieux, Daniel F. Eberl, Alan R. Kay, Robert E. Campbell, Yi Shen.

**Validation:** Sheng-Yi Wu, Nelson B. C. Serre, Yi Shen.

**Visualization:** Sheng-Yi Wu, Yurong Wen, Nelson B. C. Serre, Cathrine Charlotte Heiede Laursen, Andrea Grostøl Dietz, Abhi Aggarwal, Daniel F. Eberl, Alan R. Kay, Yi Shen.

**Writing – original draft:** Sheng-Yi Wu, Yi Shen.

**Writing – review & editing:** Sheng-Yi Wu, Yurong Wen, Nelson B. C. Serre, Cathrine Charlotte Heiede Laursen, Andrea Grostøl Dietz, Brian R. Taylor, Mikhail Drobizhev, Rosana S. Molina, Abhi Aggarwal, Vladimir Rancic, Michael Becker, Klaus Ballanyi, Kaspar Podgorski, Hajime Hirase, Maiken Nedergaard, Matyáš Fendrych, M. Joanne Lemieux, Daniel F. Eberl, Alan R. Kay, Robert E. Campbell, Yi Shen.

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
