## [Editor Report · Decision Letter 0]

14 Mar 2022

Dear Dr Shen, 

Thank you for submitting your manuscript entitled "A sensitive and specific genetically encodable biosensor for potassium ions" for consideration as a Methods and Resources article by PLOS Biology. I am sorry for the delay in getting back to you as we consulted with an academic editor about your submission. 

Your manuscript has now been evaluated by the PLOS Biology editorial staff, as well as by an academic editor with relevant expertise, and I am writing to let you know that we would like to send your submission out for external peer review.

Once your full submission is complete, your paper will undergo a series of checks in preparation for peer review. Once your manuscript has passed the checks it will be sent out for review. To provide the metadata for your submission, please Login to Editorial Manager (https://www.editorialmanager.com/pbiology) within two working days, i.e. by Mar 16 2022 11:59PM.

If your manuscript has been previously reviewed at another journal, PLOS Biology is willing to work with those reviews in order to avoid re-starting the process. Submission of the previous reviews is entirely optional and our ability to use them effectively will depend on the willingness of the previous journal to confirm the content of the reports and share the reviewer identities. Please note that we reserve the right to invite additional reviewers if we consider that additional/independent reviewers are needed, although we aim to avoid this as far as possible. In our experience, working with previous reviews does save time. 

If you would like to send previous reviewer reports to us, please email me at rhodge@plos.org to let me know, including the name of the previous journal and the manuscript ID the study was given, as well as attaching a point-by-point response to reviewers that details how you have or plan to address the reviewers' concerns. 

Given the disruptions resulting from the ongoing COVID-19 pandemic, please expect some delays in the editorial process. We apologise in advance for any inconvenience caused and will do our best to minimize impact as far as possible.

Kind regards,

Richard

Richard Hodge, PhD

Associate Editor, PLOS Biology

rhodge@plos.org

PLOS

---

## [Decision Letter · Decision Letter 1]

7 Apr 2022

Dear Dr Shen,

Thank you for submitting your manuscript "A sensitive and specific genetically encodable biosensor for potassium ions" for consideration as a Methods and Resources article at PLOS Biology. Your manuscript has been evaluated by the PLOS Biology editors, an Academic Editor with relevant expertise, and by three independent reviewers.

The reviews are attached below. You will see that the reviewers find the manuscript to be well-done and note that the biosensor would be a valuable tool for the community. However, Reviewer #1 raises several important concerns, including the accuracy of the biosensor due to its low affinity and the overall strength of the structural characterization. In addition, Reviewer #2 notes that the ratiometric behaviour of GINKO2 should be analysed in a cellular context.

In light of the reviews, we will not be able to accept the current version of the manuscript, but we would welcome re-submission of a revised version that takes into account the reviewers' comments. We cannot make any decision about publication until we have seen the revised manuscript and your response to the reviewers' comments. Your revised manuscript is also likely to be sent for further evaluation by the reviewers.

We expect to receive your revised manuscript within 3 months. Please email us (plosbiology@plos.org) if you have any questions or concerns, or would like to request an extension. At this stage, your manuscript remains formally under active consideration at our journal; please notify us by email if you do not intend to submit a revision so that we may end consideration of the manuscript at PLOS Biology.

**IMPORTANT - SUBMITTING YOUR REVISION**

*Re-submission Checklist*

*Published Peer Review*

*PLOS Data Policy*

*Blot and Gel Data Policy*

Sincerely,

Richard

Richard Hodge, PhD

Associate Editor, PLOS Biology

rhodge@plos.org

REVIEWS:

Reviewer #1: Potassium ions (K+) are essential for cellular functions across all organisms. In this manuscript, Wu et al. developed a sensitive genetically-encoded sensor to monitor K+ dynamics. Using structure-guided optimization and iterative evolution, a second generation of K+ sensor, namely GINKO2, was developed. GINKO2 shows significantly improved dynamic range, with maximum dF/F0 over 1500%. GINKO2 exhibits excitation ratiometric properties, which provides a convenient way to quantify absolute K+ concentrations across different experiments. Furthermore, GINKO2 is carefully characterized in vitro and is applied into different organisms. In general, the data quality is superb and the control experiments are well performed. The GINKO2 sensor will be a powerful addition to the family of high SNR fluorescent sensors and would help the community in general to investigate the physiological and pathological function of K+. I would support this manuscript to be accepted for publication once the following (mostly) minor issues are addressed.

1. In principle, the genetically encoded fluorescent K+ biosensors have the ability to report K+ dynamics with good spatial and temporal resolution. Unfortunately, this current manuscript falls in short in presenting clear examples of K+ dynamics within the context of good spatial resolution. For example, in Figure 5, the authors only show the expression pattern and time-series curves, yet the spatial difference of GINKO2 across different time points were missing. Similar cases can be seen in Figure 5 and Figure 6.

2. 2-photon microscopy has been widely used with fluorescent sensors to enable deeper tissue imaging in living animals. It would be good for the authors to provide info regarding 2-photon spectrum of GINKO2.

3. To image extracellular K+ dynamics in mice, GINKO2 is delivered to the brain by injection of purified GINKO2 proteins. Such a strategy is OK for imaging extracellular K+. This injection is challenging to apply compared with widely used virus-mediated expression. Have the authors tried the viral-mediated method? The authors should clarify or discuss this point.

4. In Figure 2C, the author showed the linker1 and linker2 sequence with two amino acids highlighted in blue and purple, respectively. It is not clear what it means. Figure 2B is similar to Figure1A; thus, I think it will be better to highlight the linker region in Figure 1A. An overall structure with all mutations marked could be used to help the readers to understand the possible function of other mutations that have not been discussed.

5. In Figure 1 legend, I believe the structure is for "GINKO1", which may be mistakenly labelled as "GINKO2".

6. In Figure 4, it would be good for the authors to provide a cartoon to illustrate how the experiments are performed. The similar case in Figure 7.

7. In Figure 6, the experimental design should be labeled more clearer, e.g., how was GINKO2 delivered? Additionally, the experiments were done in 2 mice? I suggested to do more repeats in different mice.

Reviewer #2: PBIOLOGY-D-22-00474R1 Wu et al.

This manuscript reports the structure and characterization of a genetically encoded potassium sensor, GINKO1, based on the insertion of the E. coli potassium binding protein Kbp into a circularly permuted version of green fluorescent protein (GFP). The structural information is used for a mutant screen to identify an improved version, GINKO2, which is then tested in a variety of systems including bacteria, plants, and mice. Although the structure is interesting, the quality of the work and presentation is serious flawed. Moreover, the fundamental question of how a sensor having a reported Kd for potassium of 15 mM, which is at least an order of magnitude below the concentration of intracellular potassium can accurately report on potassium concentration changes is never addressed in a quantiative way. This fundamental quantitative mismatch is never adequately addressed. Moreover, GINKO2 has a sensitivity to pH. Hence one is left wondering what parameter this sensor actually reports on in cellular contexts.

Despite the very high resolution data for the structure (1.85Å) the analysis of the potassium binding site is surprisingly superficial. No distances are given for the potassium coordinating carbonyl backbone ligands (Fig. 1D). This is the most critical parameter for understanding how Kdp might serve as a potassium binder. Given the unusual nature of the binding site, it is surprising that there is no comparison to other well-characterized protein based potassium binding sites (channels, Na+/K+ ATPase, for example), all of which use an apparently similar strategy in which backbone carbonyls coordinate the ion. There is also no comparison to the well studied potassium binding circular peptide valinomycin, which similarly coordinates potassium through interactions with carbonyls. The figure showing comparison with the isolated Kbp structure is poor quality. Although potassium was not located in that structure, it was determined under conditions in which the potassium binding site should be occupied. Are there meaningful changes between the crystal structure here and the Kbp structure? RMSD values are missing.

Fig. 1 also lacks a cartoon showing the design of GINKO2. Even though the design for GINKO1 has been published before, showing such a diagram is essential for understanding how and where Kbp is inserted into the circularly permuted GFP. 

Figure S1 is entirely uninformative and has poor quality data. The SEC is overloaded and provides no information about the molecular size of the protein and cannot be used to support the claim that GINKO1 is a monomer. There are no molecular weight standards. The peak shape is exceptionally broad and provides no information about the molecular size at it appears to span most of the included volume of the column. There is no reason to show the diffraction pattern. It is striking, given the claimed resolution of 1.85Å, that there are no figures showing exemplar electron density. Such information is absolutely essential for supporting the claims. Density of the putative potassium binding site should be shown .

Fig. 2 is entirely uninformative as it only shows a portion of the interface where the mutations are made. None of the context of the interactions of the linker residues is shown, making it impossible to understand how the mutations may be acting. 

The text says that the linker was randomized for the selection experiments to improve GINKO. However, the methods simply state that random mutagenesis was used. It would appear that this covers the entire protein, not the linker. The selection of clones with new properties is uncelar. E coli has 250 mM K+ in the cytoplasm (Weiden J Gen Physiol 50:1641-1661m1967). Hence, by searching for brighter GINKO variants, it is unclear exactly what he selection is based on.

The parent Kbp has a reported affinity for potassium of 160 µM. Given the structural data, is there any way to rationalize how the affinity has been lowered in the GINKO constructs?

The structure is potentially interesting. The potassium responses are unconvincing given the quantitative considerations.

Reviewer #3: Wu et al present the crystal structure of GINKO1, the first generation of a K+ biosensor. Based on the structural information they engineered GINKO2, a massively improved, ratiometric K+ sensor, which was shown to be applicable in bacteria, plant cells and in Drosophila, as well as extracellularly applied to mice. 

The data presented in the manuscript are very conclusive and GINKO2 will provide a valuable tool for studying K+ homeostasis in cells and tissues. The sensitivity and selectivity of GINKO2 is much improved when compared to GINKO1.

I have one major criticism that could easily be addressed. The ratiometric behavior of GINKO2 has only been shown for purified protein and is then used as calibration curve for in vivo experiments. It would be important to show a similar ratiometric behavior of GINKO2 in cells: One possibility would be valinomycin- and protonophore-permeabilized E. coli cell incubated at a neutral pH but with changing K+ concentrations. This way the intracellular potassium concentrations would equal the extracellular concentrations while having GINKO2 in a cellular environment.

Additionally, I have some smaller criticism:

1. The review that summarizes the role of potassium homeostasis in bacteria is a bit outdated. In recent years a lot of progress has been made with respect to bacterial potassium homeostasis, which actually better highlights the need of good K+ biosensors. I suggest to cite one of the more recent reviews or actual primary literature.

2. You state 'Notably, the backbone carbonyl oxygen atoms of six amino acids (V154, K155, A157, G222, I224, and I227) coordinate K+, similar to the coordination sphere of K+ selectivity filters in K+ channel KcsA and K+ transport protein TrkH [14].' In fact in both ion channels the potassium ion(s) in the selectivity filter are coordinated by eight not six carbonyl oxygen atoms. This might also explain the higher affinity of these proteins for potassium ions. Note that also TrkH is a potassium channel not a transporter. Also, your citation is not appropriate. The selectivity filter of TrkH is not even shown in that review. I suggest you rather cite primary literature!

3. Sentence 'This mutation may help to stabilize the K + -bound GINKO1? by reducing the distance to D148 and hence increasing their electrostatic interaction' on page 5 is somehow wrong. Why '?' and are you really referring to GINKO1 here?

4. Why would GINKO2 not respond to Na+? Any speculation based on the structure?

5. Several panels of different figures appear to lack error bars although the authors state at least triplicates have been performed.

6. Fig. S6 correct RuCl to RbCl

---

## [Editor Report · Decision Letter 2]

15 Jul 2022

Dear Dr Shen,

Thank you for your patience while we considered your revised manuscript "A sensitive and specific genetically encodable biosensor for potassium ions" for publication as a Methods and Resources article at PLOS Biology. This revised version of your manuscript has been evaluated by the PLOS Biology editors and the Academic Editor. 

Based on our Academic Editor's assessment of your revision, I am pleased to say that we are likely to accept this manuscript for publication, provided you address the following data and other policy-related requests that I have listed below (A-F):

(A) We would like to suggest the following slight modification to the title:

“A sensitive and specific genetically-encoded potassium ion biosensor for in vivo applications across the tree of life” 

(B) In the ethics statement in the Methods section, please provide the approval number issued by the Danish National Animal Experiment Committee. In addition, please confirm that this committee is an Institutional Animal Care and Use Committee (IACUC) or appropriate animal ethics committee. In addition, please provide the method of euthanasia used in the experiments.

(C) You may be aware of the PLOS Data Policy, which requires that all data be made available without restriction: http://journals.plos.org/plosbiology/s/data-availability. For more information, please also see this editorial: http://dx.doi.org/10.1371/journal.pbio.1001797

- Supplementary files (e.g., excel). Please ensure that all data files are uploaded as 'Supporting Information' and are invariably referred to (in the manuscript, figure legends, and the Description field when uploading your files) using the following format verbatim: S1 Data, S2 Data, etc. Multiple panels of a single or even several figures can be included as multiple sheets in one excel file that is saved using exactly the following convention: S1_Data.xlsx (using an underscore).

- Deposition in a publicly available repository. Please also provide the accession code or a reviewer link so that we may view your data before publication.

Regardless of the method selected, please ensure that you provide the individual numerical values that underlie the summary data displayed in the following figures, as they are essential for readers to assess your analysis and to reproduce it.

Figure 2D, 3A-I, 4B-D, 5B-C, 6D-G, 7B-E, S2, S5A-B, S6, S7, S8, S9A-C, S12A-C

(D) Please ensure that the GINKO1 structure deposited in the PDB (7VCM) is made publicly available at this stage, as it is currently on hold. 

(E) Please also ensure that each of the relevant figure legends in your manuscript include information on *WHERE THE UNDERLYING DATA CAN BE FOUND*, and ensure your supplemental data file/s has a legend.

(F) Please ensure that your Data Statement in the submission system accurately describes where your data can be found and is in final format, as it will be published as written there. Specifically, please remove the sentence saying that the data will available from the corresponding author upon request, as the data will be included in the supplementary data files. 

We expect to receive your revised manuscript within two weeks. 

*Published Peer Review History*

*Press*

Sincerely,

Richard

Richard Hodge, PhD

Associate Editor, PLOS Biology

rhodge@plos.org

PLOS

---

## [Editor Report · Decision Letter 3]

1 Aug 2022

Dear Dr Shen,

Thank you for the submission of your revised Methods and Resources article "A sensitive and specific genetically-encoded potassium ion biosensor for in vivo applications across the tree of life" for publication in PLOS Biology. On behalf of my colleagues and the Academic Editor, Raimund Dutzler, I am pleased to say that we can accept your manuscript for publication, provided you address any remaining formatting and reporting issues. These will be detailed in an email you should receive within 2-3 business days from our colleagues in the journal operations team; no action is required from you until then. Please note that we will not be able to formally accept your manuscript and schedule it for publication until you have completed any requested changes.

PRESS

Sincerely, 

Richard

Richard Hodge, PhD

Associate Editor, PLOS Biology

rhodge@plos.org

PLOS
